# Exploring the Aerosol Activation Properties in Coastal Shallow Convection Using Cloud and Particle-resolving Models

Ge Yu[1], Yueya Wang[2], Zhe Wang[2], Xiaoming Shi[2]

[1]Division of Emerging Interdisciplinary Areas, Hong Kong University of Science and Technology, Hong Kong, China
[2]Division of Environment and Sustainability, Hong Kong University of Science and Technology, Hong Kong, China

*Correspondence to*: Xiaoming Shi (shixm@ust.hk)

**Abstract.** Aerosols significantly impact the global climate by affecting the Earth's radiative balance and cloud formation. However, conducting cloud-altitude aerosol observations is currently costly and challenging, leading to gaps in accurately assessing aerosol activation properties during cloud formation. In this study, the Cloud Model 1 (CM1) is employed to investigate the movement of air parcels under shallow convection conditions in a coastal area. Subsequently, the evolution of various aerosol populations in the ideal scenarios is simulated by the PartMC-MOSAIC model to investigate their activation properties. It is found that leaving the boundary layer and entering the free atmosphere causes environmental changes in the parcels, which in turn alter the aerosol evolution and the cloud-forming potential. The impact of ascent timing is notably manifested in the concentration of ammonium nitrate rather than other chemical constituents. The rapid formation of ammonium nitrate accelerates the aerosol aging process, thereby modifying the hygroscopicity of the population. The differences between the aerosol populations in the boundary layer and at cloud altitudes highlight the necessity of vertical observations and numerical modeling. In addition, as supersaturation rises from 0.1% to 1%, the discrepancy in the CCN activation ratio between the particle-resolved results and the internal mixing assumption increases from 7% to 30%. This emphasizes the potential of appropriate mixing state parameterization in assessing aerosol activation properties. This study advances the understanding of aerosol hygroscopic changes under real weather conditions and offers insights into future modeling of aerosol-cloud microphysics.

## 1 Introduction

Atmospheric aerosols, originating from both natural and anthropogenic sources, play a critical role in the Earth's climate system by influencing radiative balance and cloud formation (Charlson et al., 1992). The interactions between aerosols and clouds remain uncertain, complicating climate research inquiries (Graf, 2004). Once emitted, aerosols undergo complex transport and aging processes, leading to significant changes in their physical and chemical properties (Calvo et al., 2013). These transformations affect their ability to act as cloud condensation nuclei (CCN), which is central to cloud formation and subsequent climate impacts (Pierce and Adams, 2009; Spracklen et al., 2011; McFiggans et al., 2006). The κ-Köhler theory (Köhler, 1936; Petters and Kreidenweis, 2007) provides a framework for understanding CCN activation, with the hygroscopicity parameter ($\kappa$) serving as a key indicator of aerosol water uptake capacity. However, ambient aerosols are rarely homogeneous; they consist of complex mixtures of inorganic ions, organic matter, black carbon, and mineral dust, which

collectively influence their hygroscopicity and CCN activity (Swietlicki et al., 2008; Zhao et al., 2020). Quantifying the cloud-forming potential remains challenging due to the spatial, temporal, and compositional complexity of aerosol populations (Rosenfeld et al., 2014).

The mixing state indicates the chemical composition distribution among aerosol particles, the treatment of which represents a crucial aspect in investigating aerosol aging processes and CCN activation properties. There are two ideal extremes of the mixing state: internally mixed and externally mixed (Winkler, 1973) commonly applied as simplified assumptions. The former assumes that each particle shares the same chemical composition, while the latter considers that each individual particle contains distinct species. In aerosol research, Riemer introduced the use of the mixing state index to quantitatively assess the

degree of mixing state (Riemer and West, 2013). Subsequently, Ching et al. (2017) provide a detailed explanation of the impact of the mixing state on CCN properties. They proved that simplified assumptions about mixing state will cause inevitable errors due to the loss of particle-level information. However, such simplifications regarding the mixing state remain inevitable, especially in experimental research and large-scale simulations.

In experimental studies, the internal mixing assumption is commonly adopted in CCN closure studies due to observational

constraints, particularly in environments with minimal freshly emitted external pollutant particles (Ervens et al., 2010; Wang et al., 2010). Aerosol parameters influencing CCN activity, such as size and chemical composition, are measured concurrently for specific samples. In field research, CCN closure studies compare predicted CCN activities with direct measurements. For instance, Juranyi et al. (2010) measured CCN number concentrations at a remote high-alpine research station. Combining dry number size distribution data from a scanning mobility particle sizer (SMPS) and bulk chemical composition data from an

aerosol mass spectrometer (AMS) and a multi-angle absorption photometer (MAAP), the closure study showed good agreement, emphasizing the importance of the mean chemical composition. However, the prediction of CCN using bulk chemical composition data does not align with the measurement in every station. Moore et al. (2012) reported large discrepancies in hygroscopic coefficients derived from size-resolved aerosol composition and size-averaged data. Simplifications in mixing state and composition could lead to 35% to 75% CCN overpredictions. In certain cases in California,

applying the externally mixed assumption or utilizing size-resolved data improved the CCN prediction. In summary, field studies have greatly enhanced our understanding of real-world aerosol populations; nevertheless, these investigations face limitations due to constraints in instrument placement and sample availability.

This necessitates the development of numerical modeling tools to investigate aerosol evolution and distribution patterns. One particle-resolving model named as Particle Monte Carlo model (PartMC) was introduced as an effective tool to address this

significant challenge (Riemer et al., 2009). PartMC is a comprehensive model that accounts for the composition and size of each single computational particle. The researchers have also coupled it with the modern Model for Simulating Aerosol Interactions and Chemistry (MOSAIC) to deterministically treat the gas-phase and gas-to-particle questions (Zaveri et al., 2008). Leveraging its particle-based approach, the PartMC-MOSAIC model can be used to explore how different mixing state assumptions influence the evolution of aerosol populations in complex environments. A series of studies employed this model

to study the related impacts of the black carbon mixing state on processes such as nucleation scavenging and cloud processing (Beeler et al., 2022; Ching et al., 2012). Tian et al. (2014) investigated the evolution of ship-emitted aerosol particles and validated the model with observations from the QUANTIFY study in 2007. Plume-exit modeling was conducted by Mena et. al (2017) to investigate the CCN activity of aerosols from residential biofuel combustion. This model has been applied in various scenarios to help interpret the CCN measurement (Razafindrambinina et al., 2023). Additionally, the model can

calculate the mixing state index defined based on the Shannon entropy (Riemer and West, 2013), facilitating studies on the significance of the mixing state in CCN activity (Ching et al., 2019; Zheng et al., 2021). Although the PartMC-MOSAIC has been extensively applied in previous studies, these studies primarily focused on parcels within the boundary layer, this may limit the accurate representation of environmental conditions when studying the cloud-forming ability of aerosols. Compared to the work mentioned above, Curtis et al. (2017) noted that PartMC, as a zero-dimensional box model, lacks spatial

information and has not been integrated into meteorological studies. To address this, they pioneered the coupling of Weather Research and Forecasting (WRF) model with PartMC, resolving aerosol composition on a per-particle level and integrating aerosol chemistry with meteorology. This represents one of the few studies to combine meteorological modeling with PartMC, which reveals the potential of PartMC in studying aerosol vertical transport. Building on this foundation, our study further explores the application of meteorological simulations coupled with PartMC, advancing the understanding of this approach in

aerosol research.

Currently, large-scale meteorological simulations typically employ simplified aerosol parameterization methods to represent the contribution of cloud condensation nuclei (CCN) to cloud microphysical processes (Thompson and Eidhammer, 2014; Morrison and Milbrandt, 2011). For instance, a common approach directly relates CCN concentration to ambient supersaturation through power-law parameterization. As proposed by Twomey (1959), the function is expressed as $N_{CCN} =$

$C \cdot S^k$, where S represents supersaturation, and C and k are coefficients derived from observed CCN characteristics, typically prescribed as parameters in simulations. Due to its simplicity, this approach remains widely used in many microphysical schemes (Hong et al., 2010; Mansell et al., 2010; Morrison et al., 2009). However, it undeniably yields coarse CCN estimates that deviate from real-world conditions. This deviation further impacts the calculation of critical properties such as cloud droplet number and droplet radius, which may affect the cloud-rain conversion processes in models (Fan et al., 2012). When

simulating a case of shallow cumulus clouds, Wang et al. (2025) corrected the basic power-law, revealing that the uncorrected power-law parameterization for CCN suppresses precipitation formation and overestimates cloud radiative cooling. Notably, the effectiveness of the correction decreases under high aerosol loading conditions. Comparative studies, such as Hazra et al. (2020), demonstrate significant variability in results across different microphysical schemes within the WRF model for the same case. The simplifications can affect CCN concentrations, potentially introducing errors in hydrometeor predictions and

impacting cloud behavior and related atmospheric processes. Therefore, the accuracy of CCN concentration is essential to reduce model uncertainties. Given the distinct characteristics of convective cloud processes across different vertical regions, the influence of vertical CCN distribution differences on precipitation is also a promising topic. Improving the estimation of

CCN variation in cloud-forming locations is crucial for enhancing cloud microphysical schemes. To this end, this study integrates the meteorological Cloud Model 1 (CM1) (Bryan and Fritsch, 2002) with the aerosol evolution model PartMC-MOSAIC for the first time. The CM1 is applied to initially investigate the movement characteristics of air parcels under a shallow cumulus convection condition. Key parameters obtained from the CM1 experiments and observations are then introduced into the PartMC-MOSAIC model to investigate the evolution of aerosol populations across different air parcel scenarios. By analyzing key indicators such as hygroscopicity, critical supersaturation, and mixing state index, this study explores the cloud-forming abilities of aerosols within different parcels.

## 2 Methodology

In this study, we designed several ideal scenarios resembling parcels in the coastal area under a typical shallow cumulus weather condition. For each parcel, the evolution of the aerosols is investigated with a detailed particle-resolved model. The relevant characteristics of the particles in different scenarios are also comprehensively analyzed to compare the cloud-forming potential of the aerosol populations.

### 2.1 Different scenarios

In coastal areas, clouds tend to form at relatively low altitudes. Under shallow cumulus convection, the ascending height of air parcels originating from near-surface levels can range from a few hundred meters to several thousand meters, reaching the altitude where cloud formation occurs. We utilized the CM1 model to perform large-eddy simulations of ideal shallow cumulus convection conditions (Siebesma et al., 2003). During the simulation, we employed tracers to track 484 parcels originating from near-surface levels, in conjunction with the other background settings. After a 6-hour simulation period, we analyzed the vertical positions of all tracked parcels. Approximately one-third (182 parcels in our simulation) of the parcels ascended to altitudes exceeding 1000 meters. Selected representative parcel trajectories are illustrated in **Supplementary Information S1**. The ascent timing exhibited a wide range of variability and was not constrained within specific time limits. These parcels usually remained at the elevated altitude (i.e. cloud altitude) for several hours after the updraft brought them into the free troposphere, while the others only exhibited height fluctuations near the surface level. The ascent speed of the parcels was relatively fast, typically taking between 8 to 20 minutes to reach altitudes exceeding 1000 meters from a near-surface height. We selected representative ascending and surface-dwelling parcels and extracted their temperature, pressure, kinetic diffusion coefficients, and other related information for subsequent scenario setups.

In **Supplementary Information S1**, the vertical distributions of some basic variables in the CM1 experiment are shown. From Fig. S1, significant environmental changes occur at altitudes above 500 meters. Clouds have already formed at the locations where ascent parcels can reach. Shallow cumulus convection is more likely to be triggered around noon when surface heat flux

is the strongest. The CM1 simulation captured several hours of this meteorological phenomenon, while the subsequent aerosol simulations using the PartMC model would span 24 hours (starting at 6 a.m.). Given the disparities between the surface level and cloud-forming altitudes, we first designed a baseline scenario where the parcels remain near the ground throughout the entire day. To align with the most probable timing for shallow convection, the parcel ascent events were scheduled around noon (12:00) with a temporal tolerance of 4 hours. The designed four scenarios for the box model are listed as follows,

Scenario A: the parcel remains near the ground.

Scenario B: the parcel ascends to an elevated altitude 2 hours after the initialization.

Scenario C: the parcel ascends to an elevated altitude 6 hours after the initialization.

Scenario D: the parcel ascends to an elevated altitude 10 hours after the initialization.

The initialization was finished after the 6-hour spin-up simulations, which is detailed described in subsection 2.2. For the parcels that stay at the surface, the background temperature was set to 26 °C (299 K), and the pressure was set to the standard atmospheric pressure. After updraft, the temperature rapidly dropped to 17 °C (290 K) and the pressure became 880 hPa, corresponding to approximately 1.2 km altitude. As the parcel ascended and remained at an elevated altitude, the emission from ground sources ceased to enter these parcels. The initial relative humidity in every scenario was set to 50% and some other environmental parameter configurations for different scenarios are listed in Table 1. Besides, the horizontal eddy diffusivity at the elevated altitude was about one-fifth of the value near the surface.

**Table 1. Fundamental settings for 4 different scenarios.**

| Scenario | Ascent Timing (h) | Temperature before Ascent (K) | Temperature after Ascent (K) | Pressure before Ascent (hPa) | Pressure after Ascent (hPa) | Initial Relative Humidity (%) |
|----------|-------------------|-------------------------------|------------------------------|------------------------------|-----------------------------|-------------------------------|
| A | - | 299 | - | 1000 | - | 50 |
| B | 2 | 299 | 290 | 1000 | 880 | 50 |
| C | 6 | 299 | 290 | 1000 | 880 | 50 |
| D | 10 | 299 | 290 | 1000 | 880 | 50 |

After obtaining the updraft-related settings from the 6-hour CM1 results, we consulted some other sources to refine the long-time scenario configuration. The background conditions including the diurnal changes in mixing height and the emission data were derived from the idealized plume scenario settings conducted by Riemer et al. (Riemer et al., 2009). The aerosol data at the surface was obtained from the observations at a general air quality monitoring station in Hong Kong (Wang and Yu, 2017). The nitrate levels observed in Hong Kong (used in this study) may be higher than in other coastal cities due to historical

pollution patterns from a decade ago, but the methodology can be adapted to incorporate different observational data from other regions. The observational data provided the mass concentrations and the lognormal distribution fitting parameters of different species (mass median aerodynamic diameter MMAD, and geometric standard deviation GSD) for three modes. In the initial aerosol input for PartMC, the particles in each mode were assumed to be internally mixed, requiring the input of the

particle geometric mean diameter ($D_{gm}$) and the logarithm (base 10) of the geometric standard deviation of the diameter ($\sigma_g$). We converted mass concentrations into number concentrations and transformed the MMAD into the geometric mean diameter (GMD) for the calculations. Since the GMD and GSD of the species within each mode were relatively similar, the required ground-level background input parameters were derived using a number concentration-weighted averaging method. For cloud-altitude aerosol background conditions, we extracted the reanalysis aerosol data of Modern-Era Retrospective Analysis for

Research and Applications, Version 2 (MERRA-2, DOI: 10.5067/LTVB4GPCOTK2) in the Hong Kong region. The data of the lowest atmospheric layers (from 880 to 1000 hPa) in summer was processed and the vertical proportionality relationships between the ground-level aerosol species and cloud-altitude species were estimated. Subsequent calculations determined the mass concentrations of different species at elevated altitudes. The same size distribution and mixing state assumptions as those used for ground-level conditions were applied, and the cloud-altitude background aerosol for the simulation was ultimately

generated. The substances in the ascent parcels would dilute with cloud-altitude background aerosols and gases. Detailed information on the number concentration and size distribution of the aerosols in different modes can be found in Table 2. The initial aerosol conditions in the parcels contain three modes that are the same as the Ground part because all the parcels stay near the surface during the initialization period. The concentrations for background and emission gas are listed in **Supplementary Information S5**.

**Table 2. Aerosol conditions for the emissions (Riemer et al., 2009) and different backgrounds.**

| | N (m$^{-3}$) | $D_{gm}$ (µm) | $\sigma_g$ | Composition by Mass |
|---|---|---|---|---|
| **Emissions** | | | | |
| Meat cooking | $9 \times 10^6$ | 0.0865 | 1.9 | 100% OC |
| Diesel vehicles | $1.6 \times 10^8$ | 0.05 | 1.7 | 30% OC, 70% BC |
| Gasoline vehicles | $5 \times 10^7$ | 0.05 | 1.7 | 80% OC, 20%BC |
| | | | | |
| **Ground-level** | | | | |
| Condensation | $1.26 \times 10^9$ | 0.0816 | 1.6 | 32.6% BC, 48.8% OC, 14% SO$_4$, 5.6% NH$_4$ |
| Droplet | $2.68 \times 10^8$ | 0.14 | 2.1 | 26.4% BC, 23.8% OC, 37.4% SO$_4$, 11.2% NH$_4$, 0.4% Cl, 0.7% NO$_3$ |
| Coarse | $2.3 \times 10^5$ | 2.16 | 1.65 | 11.8% BC, 21% OC, 12.2% SO$_4$, 0.5% NH$_4$, 17.8% Cl, 18.6% NO$_3$, 18.1% Na |
| | | | | |
| **Cloud-altitude** | | | | |
| Condensation | $3.1 \times 10^8$ | 0.0831 | 1.66 | 15.8% BC, 44.5% OC, 29.8% SO$_4$, 9.9% NH$_4$ |
| Droplet | $6.8 \times 10^7$ | 0.16 | 2.05 | 9.1% BC, 15.5% OC, 57.1% SO$_4$, 17.2% NH$_4$, 0.1% Cl, 1% NO$_3$ |

| | | | | |
|---|---|---|---|---|
| Coarse | $6.6 \times 10^4$ | 2.16 | 1.64 | 5.7% BC, 19.1% OC, 25.8% SO$_4$, 1% NH$_4$, 4.4% Cl, 39.6% NO$_3$, 4.4% Na |

## 2.2 PartMC simulations

The coupled PartMC-MOSAIC model was applied in this study to simulate the evolution of individual aerosols in a Lagrangian air parcel (Riemer et al., 2009; Zaveri et al., 2008). PartMC is a 0-D box model that resolves the specific size and composition of the remaining particles within the computational volume. The volume used is about a few cubic centimeters and well mixed, which represents a larger parcel suspended in specific scenarios. Instead of tracking the physical positions, the model focuses on monitoring the number, mass, and full composition distributions of the simulated particles (Zaveri et al., 2010). When the parcel leaves the exhaust source, no freshly emitted particles and gas will be added to the computational volume. Aside from the emission module, the number concentration of the particles is also impacted by the dilution, and coagulation processes that are stochastically calculated in the PartMC model. Within the box model, the meteorological conditions and gas concentrations are assumed to be homogeneous. The PartMC model incorporates a diffusion algorithm to simulate material exchange between the Lagrangian parcel and its surrounding environment, handling both gas and aerosol particle exchanges. Additionally, the algorithm separates horizontal and vertical diffusion coefficients. The vertical diffusion rate depends on both the magnitude and the variation rate of mixing height, with an increase in the mixing height leading to positive values. The horizontal coefficient requires user-defined input and represents the effects of horizontal turbulent diffusion. The diurnal variation of the mixing height and the surface-level horizontal coefficient in this study is consistent with Riemer et al. (2009). When the parcel is at ground level, the horizontal diffusion coefficient is set as $1.5 \times 10^{-5}$ s$^{-1}$. After the parcel rises, the value of mixing height is constant to achieve zero vertical diffusion rate, simulating free tropospheric conditions. The horizontal coefficient at the elevated altitude is set as $3 \times 10^{-6}$ s$^{-1}$, corresponding to the one-fifth ratio relative to the surface coefficient as described in subsection 2.1. The temporal evolution of horizontal coefficients for Scenarios A~D is documented in **Supplementary Information S5**. When coupled with the MOSAIC model, some other processes such as condensation, evaporation, and chemical reactions can be included in the simulation. MOSAIC consists of modules for gas-phase photochemistry (Zaveri and Peters, 1999), particle-phase thermodynamics (Zaveri et al., 2005), and gas-particle mass transfer (Zaveri et al., 2008). With these modules, the model could treat a total of 77 gas species and the atmospherically important aerosol species like sulfate (SO$_4$), nitrate (NO$_3$), chloride (Cl), carbonate (CO$_3$), ammonium (NH$_4$), sodium (Na), calcium (Ca), methane sulfonic acid (MSA), black carbon (BC), primary organic aerosol (OC), and several secondary organic aerosol (SOA) species. A more detailed description of the governing equations and numerical methods used in the model is given in Riemer et al. (2009).

We used PartMC version 2.6.1 for this study and initialized the number of computational particles as $10^5$. The time step of the simulation was 60 seconds. The simulations were conducted for 30 hours. The first 6-hour simulation (from 0:00 a.m. to 6:00 a.m.) is only used for the spin-up process, and the subsequent 24-hour simulation (from 6:00 a.m. to 6:00 a.m. on the next day) is employed to analyze the diurnal variation of the aerosols.

## 2.3 CCN activity computation

To quantitatively assess the cloud-forming potential of an aerosol population, we could investigate the CCN activity at a specific supersaturation level. This factor is a straightforward proportion between activatable CCN particles and the total aerosol particles (CN, condensation nuclei). PartMC-MOSAIC could provide useful information about each particle, allowing us to calculate the hygroscopicity value ($\kappa_i$) and subsequently determine the critical supersaturation ($S_{c,i}$) required for activation. Hence, we could evaluate whether each particle is activated at a certain supersaturation with the $S_{c,i}$ and get the CCN/CN ratio (CCN activity).

The hygroscopicity parameter $\kappa_i$ of the particle $i$, as introduced by Ghan et al. (2001), and Petters and Kreidenweis (2007), is a dimensionless parameter to reflect the water uptake property of the aerosol. The water activity $a_{w,i}$ of the particle $i$ is given by,

$$\frac{1}{a_{w,i}} = 1 + \kappa_i \frac{V_{dry,i}}{V_{w,i}} \tag{1}$$

where $V_{dry,i}$ is the dry particle volume and $V_{w,i}$ is the volume of water in the particle. For the particle $i$ containing various components, $\kappa_i$ is the volume-weighted average of the $\kappa$ values of its constituent aerosol species. For the aerosol population, the median value $\kappa_m$ of all $\kappa_i$ values is calculated to serve as a characteristic parameter of the ensemble hygroscopicity, which is analyzed in Section 3. Table 3 lists specific $\kappa$ values assigned to different species simulated in this study (Clegg et al., 1998; Riemer et al., 2009; Zaveri et al., 2010).

The equilibrium saturation ratio $S_i(D_i)$ over an aqueous droplet is given by the following Köhler equation:

$$S_i(D_i) = a_{w,i} \exp\left(\frac{4\sigma_w M_w}{RT\rho_w D_i}\right) \tag{2}$$

where $\sigma_w$ is the surface tension of the water-air interface, $M_w$ is the molecular weight of the water, $R$ is the universal gas constant, $T$ is the temperature, $\rho_w$ is the density of water, and $D_i$ is the wet diameter of the particle. Combining the equations and using the $D_i$ and the dry diameter $D_{dry,i}$ to replace the respective volumes, we obtain the $\kappa$-Köhler equation introduced by Petters and Kreidenweis (2007),

$$S_i(D_i) = \frac{D_i^3 - D_{dry,i}^3}{D_i^3 - D_{dry,i}^3(1 - \kappa_i)} \exp\left(\frac{4\sigma_w M_w}{RT\rho_w D_i}\right) \tag{3}$$

We could compute the critical wet diameter $D_{c,i}$ first and then calculate the critical supersaturation ratio $S_{c,i}$. We may set the $\partial S_i(D_i)/\partial D_i$ to zero to numerically solve the $D_{c,i}$ at which $S_i(D_i)$ is maximal. Substituting the $D_{c,i}$ into the equation and $S_{c,i}$

equals $(S_i(D_{c,i}) - 1)$ percent. Once the $S_{c,i}$ is equal to or less than the environmental supersaturation, the CCN-active aerosols could be identified and counted.

**Table 3. Hygroscopicity parameters were assigned to the species used in the study.**

| Aerosol species | $\kappa$ |
|---|---|
| $NO_3$ | 0.65 |
| $SO_4$ | 0.65 |
| $NH_4$ | 0.65 |
| Na | 1.1 |
| Cl | 1.1 |
| BC | 0 |
| OC | 0.001 |
| SOA | 0.1 |

**3 Results for the aerosol activation properties**

**3.1 The changes in the aerosol chemical compositions**

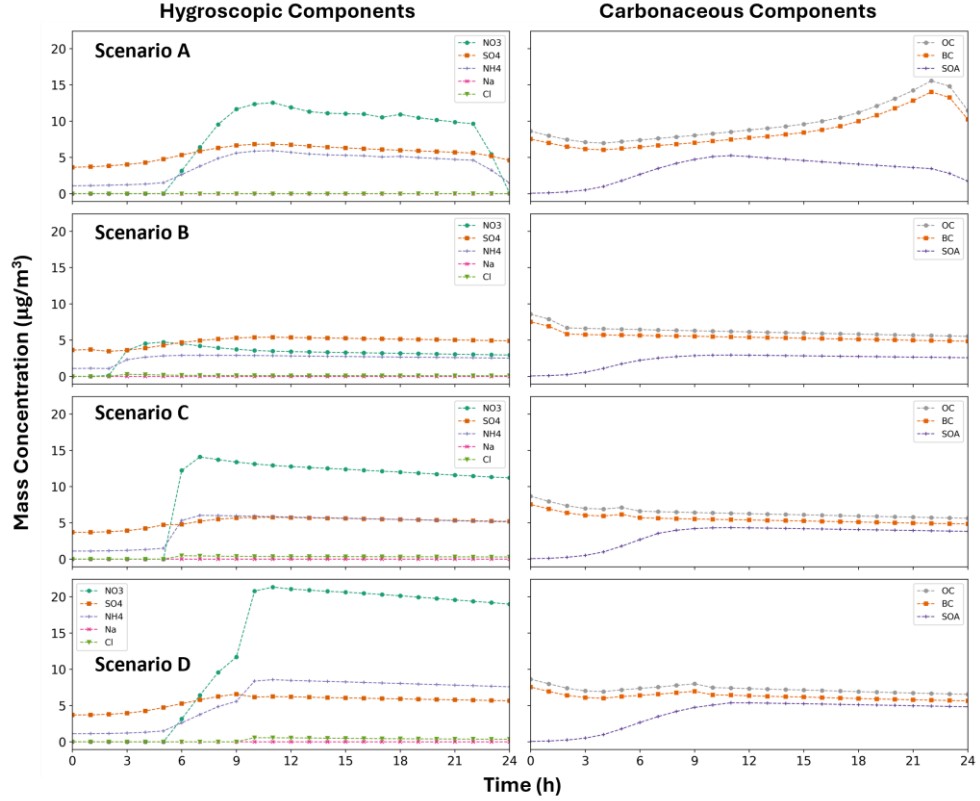

**Figure 1. Mass concentration variations over time of hygroscopic and carbonaceous components in Scenario A, B, C, and D. For Scenario A: the parcel remains near the ground; B, C and D: the parcel ascends after 2 hours, 6 hours and 10 hours respectively.**

Starting from this section, we analyze the results from the PartMC-MOSAIC simulations. The aerosol data of the PartMC parcels representing the four scenarios are extracted and plotted separately. To evaluate the sources, transport, and transformations of the substances in these aerosol populations, the mass concentration variations of the hygroscopic and carbonaceous components in aerosols are investigated, as shown in Fig. 1. In the coastal area, the main components of hygroscopic aerosols include sulfate, nitrate, ammonium ions, and sea salt, which corresponds to relatively higher

hygroscopicity parameters $\kappa$. Among these components, the concentration changes of ammonium and nitrate are the most pronounced across different scenarios. The mass concentration of sulfate generally hovers around 5 µg/m³. By contrast, the content of sodium and chloride is relatively low, not exceeding 1 µg/m³ in all simulation cases. Specifically, the parcel containing aerosols in Scenario A remains below the boundary layer, where temperature and humidity are stable. As mentioned in the methodology, the time analyzed throughout this section starts at 6:00 a.m. ($0^{th}$ hour). Therefore, after 9:00 a.m. (the $3^{rd}$

hour in the figures), the influence of sunlight and anthropogenic emissions leads to a significant increase in gas concentrations. The concentrations of ammonia ($NH_3$) and nitric acid ($HNO_3$) near the ground peak around the $5^{th}$ hour due to gas emissions and photochemical reactions, facilitating the conversion to ammonium nitrate ($NH_4NO_3$) and promoting the growth of new particles. After the $21^{st}$ hour, the combination of decreased emission and enhanced dilution accounts for the reduction in the mass concentration of the aerosols. In Scenarios B, C, and D, the parcel rises at the $2^{nd}$, $6^{th}$, and $10^{th}$ hour respectively,

corresponding to the rapid decrease in the temperature and pressure. According to the equilibrium reactions introduced in the MOSAIC model, the gas-particle transfer processes for nitrate and ammonium ions are sensitive to temperature changes, leading to a rapid increase in nitrate and ammonium ion concentrations. Specifically, in Scenario B, the early ascent brings the parcel to a relatively clean environment before sunlight and anthropogenic factors significantly influence gas substances. Hence, the concentration of the ammonium nitrate does not exhibit the same pronounced changes as in the other two scenarios.

The detailed and clear mass changes during the two hours around the parcel ascent are shown in **Supplementary Information S2**.

The emissions cease after the air parcel leaves the boundary layer and reaches higher altitudes. However, the diffusion effect persists although it weakens. Due to the absence of fresh precursors and the background with lower aerosol concentration, the concentrations of ammonium and nitrate ions gradually decline. Meanwhile, as the duration of emissions increases for parcels

that stay in the boundary layer for a longer time, the peak concentration of ammonium nitrate significantly rises. Specifically, the peak concentration of ammonium nitrate in Scenario D is approximately 4 times that of Scenario B.

Carbonaceous components include BC, OC, and SOA. They exhibit lower hygroscopicity ($\kappa$) values compared to hygroscopic components. In Scenario A, the mass concentration of OC and BC steadily increases due to the accumulation of urban emissions from the $3^{rd}$ to $21^{st}$ hour, as seen in Fig. 1. Their concentrations start to decline after the $21^{st}$ hour due to decreased

emissions and increased dilution. Differently, the mass concentration of SOA decreases after the $11^{th}$ hour because of the

cessation of the transformation from OC to SOA. When the parcel rises above the boundary layer, the concentrations of OC and BC are predominantly influenced by the dilution module rather than emission sources as seen in Scenarios B, C, and D. The formation of SOA is primarily driven by light exposure, exhibiting a variation pattern generally consistent with Scenario A. Besides, the specific peak concentrations of SOA increase as the emission accumulates.

## 3.2 The evolution of hygroscopicity in different scenarios

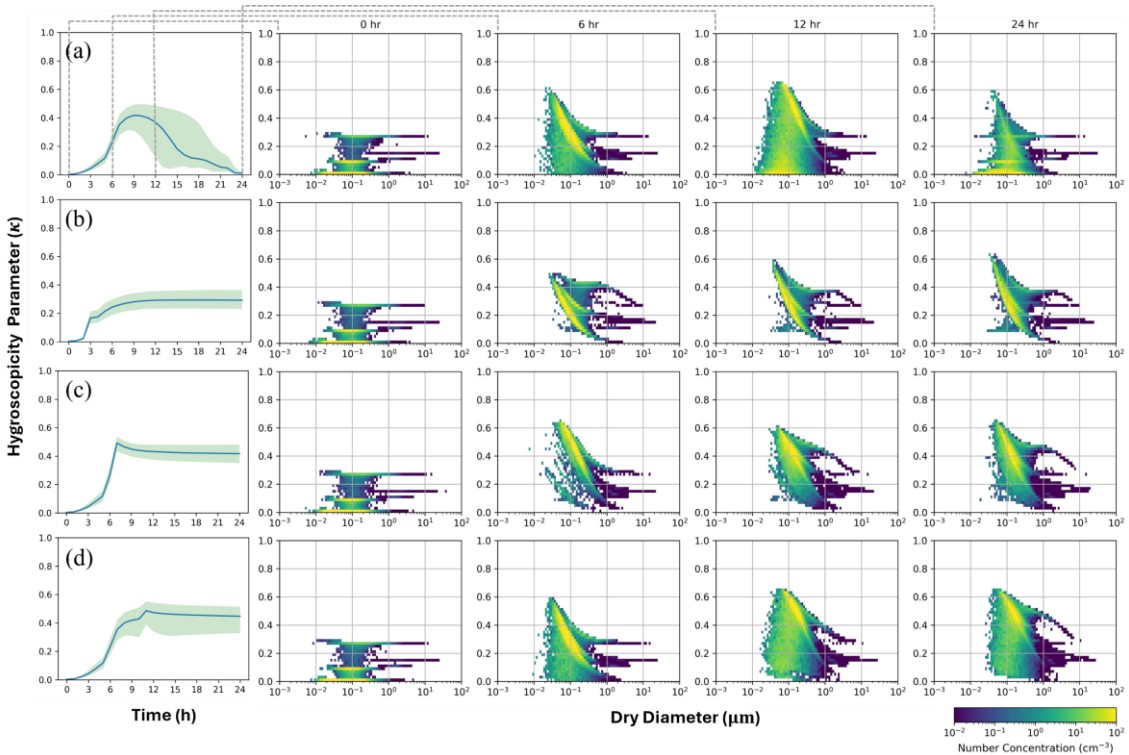

**Figure 2. The median hygroscopicity parameter $\kappa_m$ changes over time and the number concentration distribution of the particles with different $\kappa_i$ values across particle size at 0, 6, 12, and 24 hours in (a) Scenario A, (b) Scenario B, (c) Scenario C, and (d) Scenario D. The blue solid line in the left plots represents the median $\kappa$ across the aerosol population, while the edges of the green shading**
**denote the 25th and 75th percentile $\kappa_i$ values.**

Figure 2 illustrates the evolution of the median hygroscopicity parameter $\kappa_m$ over time and the selected particle-resolved hygroscopicity distribution versus particle size in four different scenarios. In Scenario A, the median $\kappa_m$ value rises during the initial 9 hours and then gradually declines over the subsequent 15 hours. Such trends align closely with the variation of the hygroscopic components, especially sulfate, nitrate, and ammonium. By contrast, in Scenarios B, C, and D, the ascent of the

parcel corresponds to an increase in the median value $\kappa_m$. This is because the updraft process results in a rapid decrease in the temperature and pressure, which speeds up the gas-to-particle conversion process, thereby accelerating the formation of hygroscopic chemical components. In addition, due to the absence of an emission term and the reduced diffusion effect, the median $\kappa$ and the overall distribution remain relatively stable after the parcel ascends to elevated altitudes.

Aside from the hygroscopicity parameter $\kappa_i$, the particle size also influences the activation properties of aerosols as described in section 2.3. Thus, the number concentration distribution of the particles with different $\kappa_i$ values across particle size is also depicted in the subplots of Fig. 2. The temporal changes of the aerosol size distribution under different scenarios are directly shown in **Supplementary Information S3**. From Fig. S4, the average size of the aerosol populations becomes larger with time owing to the condensation and coagulation processes. This aging process of the aerosols could also be seen from the subplots in Fig. 2, where the yellow portion representing high-concentration aerosols shifts towards larger particle sizes.

Regarding the specific hygroscopicity distribution with the particle size, it is evident that all the scenarios share a similar pattern at the very beginning of the simulation shown in the second column of Fig. 2. The bright yellow part predominates the population because of the large number concentration. On the $0^{th}$ hour, there are three bright lines parallel to the x-axis representing the three sub-populations of the initial aerosol conditions mentioned in section 2.1. The sub-population at $\kappa \approx 0$ comes from fresh emissions mainly consisting of the black carbon and the primary organic carbon. The other two sub-populations at hygroscopicity around 0.1 and 0.3 bring the features of the background mixture containing ammonium, sulfate, nitrate, sea salt, aged black carbon, and organic carbon. As time passes by, the sub-populations characterized by bright yellow have exhibited an increased hygroscopicity, which is attributed to the aging process of black carbon and the formation of ammonium nitrate. This phenomenon is evident in the third column, which depicts the distribution for the $6^{th}$ hour. In Scenario B, the parcel has been stationed at a cloud altitude after the $2^{nd}$ hour and shows a lower aging level compared to others. Conversely, in Scenario C, the ascent process of the parcel has accelerated the aging process, thus shifting the predominant sub-population to a higher $\kappa$ distribution by the $6^{th}$ hour. Pronounced differences emerged between Scenarios B and C following parcel ascent. This is attributed to the earlier ascent timing in Scenario B, where fewer accumulated precursor gases (e.g., ammonium nitrate precursors) were entrained. Photochemical reactions involving ozone ($O_3$) depend on solar radiation and could affect interactions between nitrogen oxides ($NO_x$). The limited time leads to lower nitric acid levels in Scenario B before ascent compared to Scenario C. As shown in Figure 1 for Scenario A, ground-level pollutant emissions caused rapid nitrate aerosol growth after the $5^{th}$ hour, preceding the ascent timing in Scenario B. Subsequently, the aerosol populations in Scenarios B and C retained their distinct characteristics but underwent similar aging trajectories. On the $12^{th}$ hour, the aerosols in Scenario D retain many original features after the parcel rises. However, in Scenario A, a significant proportion is occupied by freshly emitted hydrophobic particles. After sunset, it becomes difficult for accumulated pollutants to disperse. Furthermore, the aging process has also slowed down owing to reduced photochemical reactions. Over the last 3 hours, the sub-populations with high hygroscopicity have gradually diminished in Scenario A. This can be attributed to an intensified diffusion effect resulting from the changes in the mixing height.

**3.3 Critical supersaturation distribution and CCN activation ratio**

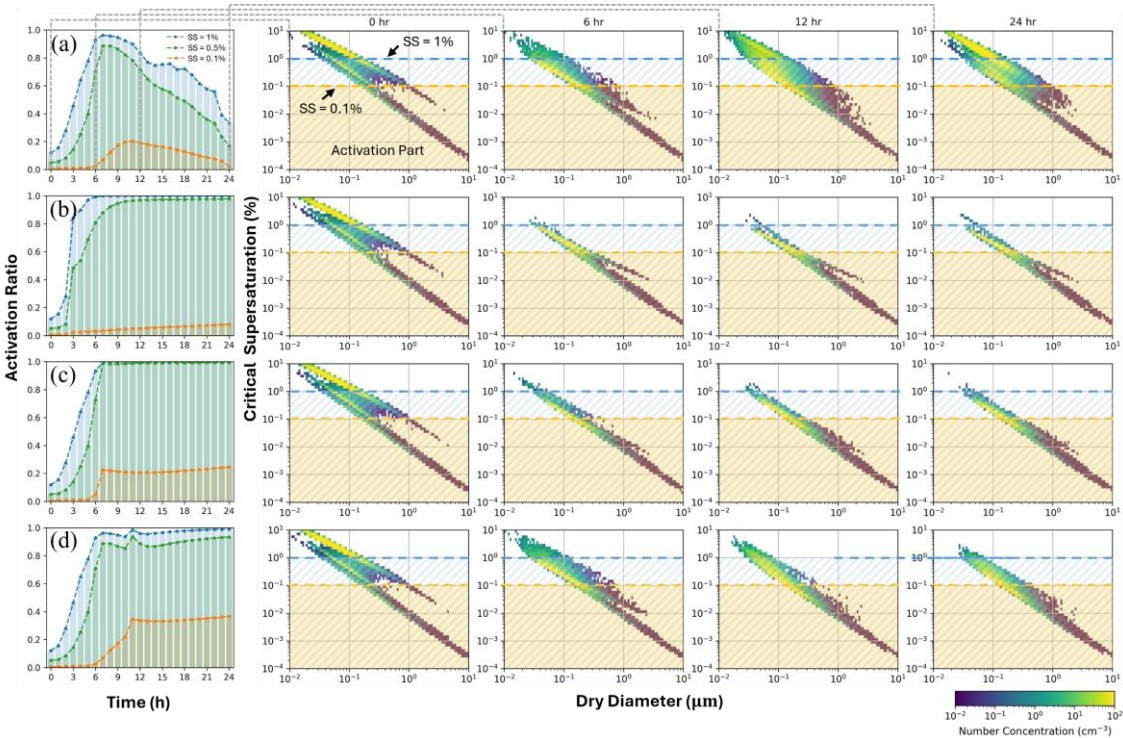

Figure 3. The temporal variations of CCN activity at 0.1%, 0.5%, and 1% supersaturation, as well as the critical supersaturation distribution across particle size at 0, 6, 12, and 24 hours in (a) Scenario A, (b) Scenario B, (c) Scenario C, and (d) Scenario D. The orange and blue horizontal dashed lines indicate supersaturations equaling 0.1% and 1%, respectively.

The right part of Fig. 3 shows the distribution of critical supersaturation ($S_c$) across particle dry diameter, at the same selected times as Fig. 2. As described in the methodology section, the critical supersaturation of each particle is a fundamental property that directly relates to its ability to become activated as a CCN. The horizontal dashed lines represent the supersaturation values equaling 0.1% and 1%. Below the lines are the particles that could be activated at this supersaturation and the color bar represents the number concentration. The value of $S_{c,i}$ depends on the hygroscopic characteristic decided by all the mixed species and the dry diameter. When the diameter becomes larger, the overall trend of the critical supersaturation should become lower as per the Köhler formula. The dry diameter is not the only impacting factor, so the distribution of this value is not concentrated along a single line in the plot. The spread pattern is also largely affected by the heterogeneity of $\kappa_i$, which is determined by the chemical composition of the particles. The sensitive impacts of different compositions on the overall $S_c$ are studied in **Supplementary Information S4**. As observed in Figures S5 and S6, some hygroscopic species tend to distribute on smaller particles during the aging process, necessitating consideration of the specific mixing state when evaluating the activation properties of particles.

At the very beginning, the particles at 0.1 μm have the $S_{c,i}$ ranging from 0.2% to 2%, and the $S_{c,i}$ of most of them is higher than 1%. Besides, a large number of particles is smaller than 0.1 $\mu m$ at this period. Therefore, the aerosol population exhibits a low activation potential since the supersaturation levels during cloud formation typically remain below 1%. At the 6$^{th}$ hour, the yellow portion in Scenario A and D has shifted to the lower part of the subplots. This is attributed to the overall increase in particle size and $\kappa_i$ values observed in the majority of the particles. The range of $S_{c,i}$ for most particles is between 0.1% and

1%, significantly influencing the activation ratio of the aerosol population within this range of supersaturation. In Scenario B and C, the earlier updraft process results in lower $S_c$ levels for the bright yellow sub-populations. By the 12$^{th}$ hour, the characteristics of bright areas in scenarios B, C, and D, like the ones at the 6$^{th}$ hour, are largely preserved on the graph. The colored regions have reduced in size because of the dilution module. However, a significant portion of high-$S_{c,i}$ and small-size particles from the freshly emitted aerosols remain within the parcel of Scenario A. After sunset, the dilution process near the

ground weakens, leading to a pronounced accumulation of pollutants. Until the 20$^{th}$ hour, the similar process continues. Combined with the diminished chemical conversion process caused by changed solar conditions, the high critical supersaturation of most aerosols in Scenario A is noticeable at the 24$^{th}$ hour. At this time, the differences between parcels in three other scenarios and the parcel near the surface become more evident. The CCN activity either remains high or rapidly increases after ascending in the latter three scenarios. It is implied that these scenarios exhibit an enhancement in the ability of

aerosols to act as CCN compared to the surface scenario after the 12$^{th}$ hour (in the nighttime).

The left column subplots of Fig. 3 compare the particle-resolved CCN activation ratio at 0.1%, 0.5%, and 1% supersaturation along with time between four scenarios. The gray dashed lines between the subplots represent the time-corresponding results. The colored curves on the left visually demonstrate the statistical outcomes that arise from the details of the $S_c$ distribution presented on the right. Moreover, the trends observed in the colored curves for each scenario are consistent with the median $\kappa$

depicted in Fig. 2.

## 4 The impact of mixing state on CCN activation

### 4.1 Relationship of error in CCN calculation and mixing state index

The mixing state index ($\chi$) is defined by Riemer and West (2013) to quantify the aerosol mixing state, which is calculated by,

$$\chi = \frac{D_\alpha - 1}{D_\gamma - 1} \tag{4}$$

where $D_\alpha$ represents the average per-particle species diversity, $D_\gamma$ represents the bulk population species diversity. Given the focus on analyzing CCN properties, the aerosol species are classified into three distinct categories based on their hygroscopicity: low-hygroscopicity species (BC, OC), high-hygroscopicity species (SO4, NO3, NH4, Cl, Na), and intermediate-hygroscopicity species (SOA). As a result, the value of $D_\alpha$ and $D_\gamma$ ranges from 1 to 3. In the state of fully external mixing ($\chi = 0$), each particle within the population possesses a distinct chemical composition. As $\chi$ increases, the mixing state gradually shifts toward internal mixing. Specifically, when $\chi$ equals 1, the aerosol population transitions entirely to a fully internal mixing state, in which all particles possess identical mass compositions.

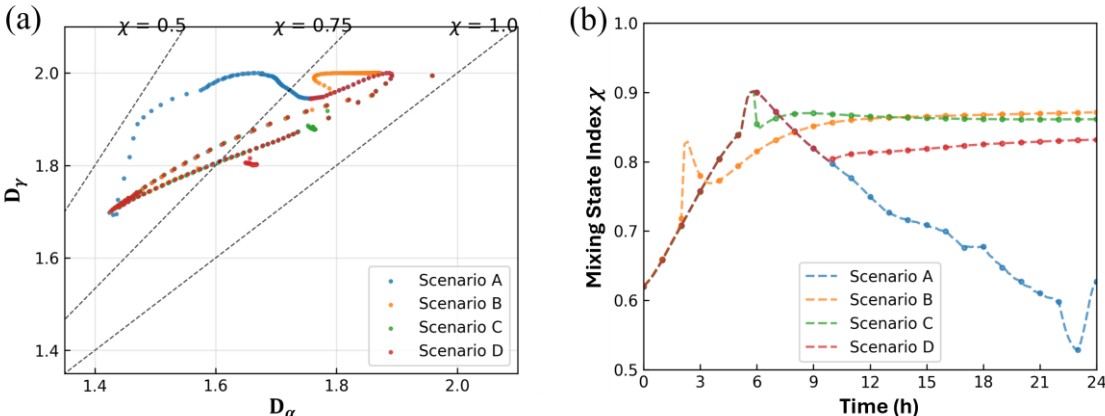

**Figure 4. (a) Scatterplot showing the definition of the mixing state index $\chi$ for four scenarios. The scatters show the state of the average per-particle species diversity $D_\alpha$ and the bulk population species diversity $D_\gamma$ at ten-minute intervals. (b) Temporal changes of the mixing state index $\chi$ for four scenarios. Every scatter represents the value for each hour.**

Figure 4a shows the scatterplot of the average per-particle species diversity $D_\alpha$ and the bulk population species diversity $D_\gamma$ at different times for various scenarios. It can be seen that the background aerosols are typically in a well-internally mixed state. Because of the different compositions, the introduction of primary emissions may cause a decrease in the $\chi$ value. Besides, the process of physical or chemical aging naturally leads to an increase in the $\chi$ value of the aerosol population. Figure 4a presents the distribution of $\chi$ at 10-minute intervals and all the samples have a large value indicating more internally mixed. The $\chi$ value for the samples ranges from 0.5 to 1. In Scenario A, there is a larger amount of freshly emitted particles compared to the other scenarios due to the prolonged influx of emissions. Without the aging process, their compositions differ from the

characteristics of the particles experiencing a lot of natural processes. As a result, some samples in Scenario A could exhibit a
more externally mixing state and the $\chi$ value is lower than the other ones.

Figure 4b depicts the changes in the mixing state index of the four aerosol populations within 24 hours. It is evident that in the three scenarios where the parcel undergoes ascent, the $\chi$ value deviates from the curve in Scenario A after ascending. Notably, the aerosol population with lower pollution accumulation demonstrates an increasing trend in the $\chi$ value during the ascent, whereas the other two scenarios do not show a significant increase in the mixing state index. The observed trend in Scenario
B highlights the influence of aging level on the $\chi$ value. The deeper the aging level of the total population, the larger the mixing state index. To a certain extent, the changes in the $\chi$ value exhibit similar characteristics to the curves of the median $\kappa$ value for every scenario. The primary emissions, dilution, chemical reactions, and other aging processes may have a similar impact on both the $\chi$ and $\kappa$ values. All these processes can alter the specific chemical composition of particles, which in turn affects the general mixing state and hygroscopicity.

Due to measurement limitations on aerosol components, the hygroscopicity of individual particles is often estimated using the average chemical composition of the total aerosol population, which is based on the fully internal mixing assumption. Applying this method we calculate a new CCN concentration $N_{CCN,avg}$, which has a discrepancy compared to the $N_{CCN}$ achieved from the particle-resolved results. The total particle concentration is defined as $N_{total}$, and the difference can be calculated by the following equation,

$$Difference\ in\ CCN\ Activation\ Ratio = \frac{|N_{CCN,avg} - N_{CCN}|}{N_{total}} \qquad (5)$$

The internal mixing assumption simplifies the calculations and the measurements and is widely used to investigate CCN activation. However, the real-world mixing state of aerosols could largely deviate from this assumption, which means the actual $\chi$ value is lower than 1. Since only particle-resolved methods accurately represent the true mixing state of aerosols, we use the CCN activation error to represent the difference between the composition-averaged and particle-resolved methods.


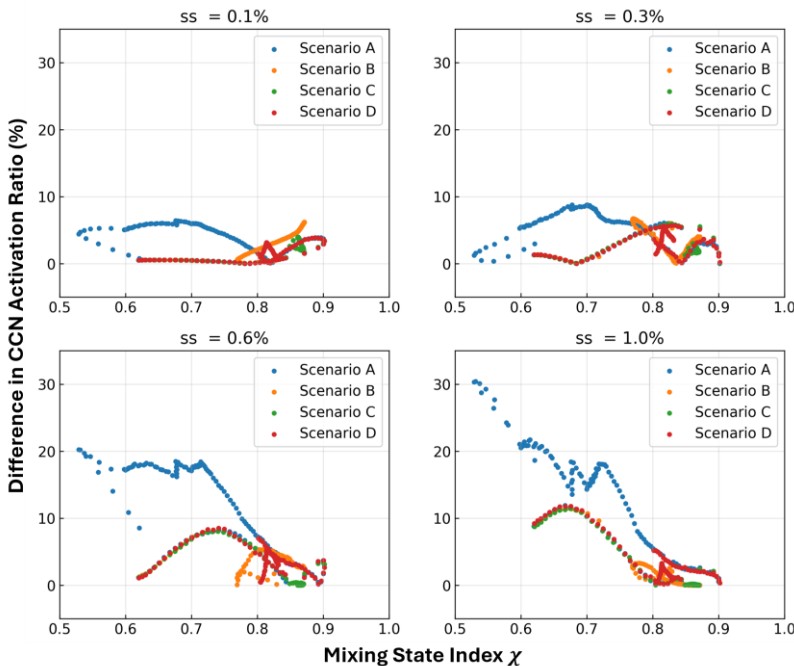

**Figure 5. The difference in CCN activation between composition-averaged and particle-resolved results with the mixing state index χ at various environmental supersaturations for four scenarios. The scatters show the results at ten-minute intervals.**

As shown in Figure 5, the CCN activation differences between the composition-averaged and particle-resolved method with the variation of the mixing state index at different environmental supersaturations are investigated. While retaining their original size distributions, particles were assigned identical chemical compositions and hygroscopicity parameters.

When the supersaturation levels are 0.1% and 0.3%, the average error in CCN activation caused by composition averaging is less than 0.1. However, as the supersaturation levels increase to 0.6% and 1%, the CCN activation errors in some scenario samples may become larger. Specifically, in Scenario A, this phenomenon is more pronounced, and smaller χ values correspond to larger error values. This is because, under higher supersaturation conditions, smaller particles within the population exhibit a greater likelihood of activation. In Scenario A, these smaller particles are primarily composed of freshly emitted pollutants with lower hygroscopicity. Consequently, even under elevated supersaturation levels, this sub-population remains inactivated in the real world. PartMC can accurately capture the mixing state reflected by a smaller mixing state index χ, and correctly calculate the critical supersaturation of these particles. In contrast, the composition-averaged method overestimates the hygroscopicity of these small, freshly emitted particles. This overestimation leads to an elevated critical supersaturation, making these particles more likely to be assumed as CCN under higher supersaturation conditions. As a result, the CCN predictions based on the internal mixing assumption become more sensitive to variations in environmental supersaturation. Overall, when χ is less than 0.7, the errors at a supersaturation level of 1% tend to be larger than those at a supersaturation level of 0.6%. This highlights the limitations of the composition-averaged method, particularly in scenarios

dominated by freshly emitted, less hygroscopic particles. In such cases, the aerosol population may exhibit a more non-uniform or externally mixed state, where chemical components can exhibit significant differences among particles. It is important to be noted that the assumption of internal mixing for the whole population can lead to absolute CCN activity errors exceeding 30% when the mixing state is not as homogeneous.

## 4.2 CCN calculation error analysis from a size-resolved perspective

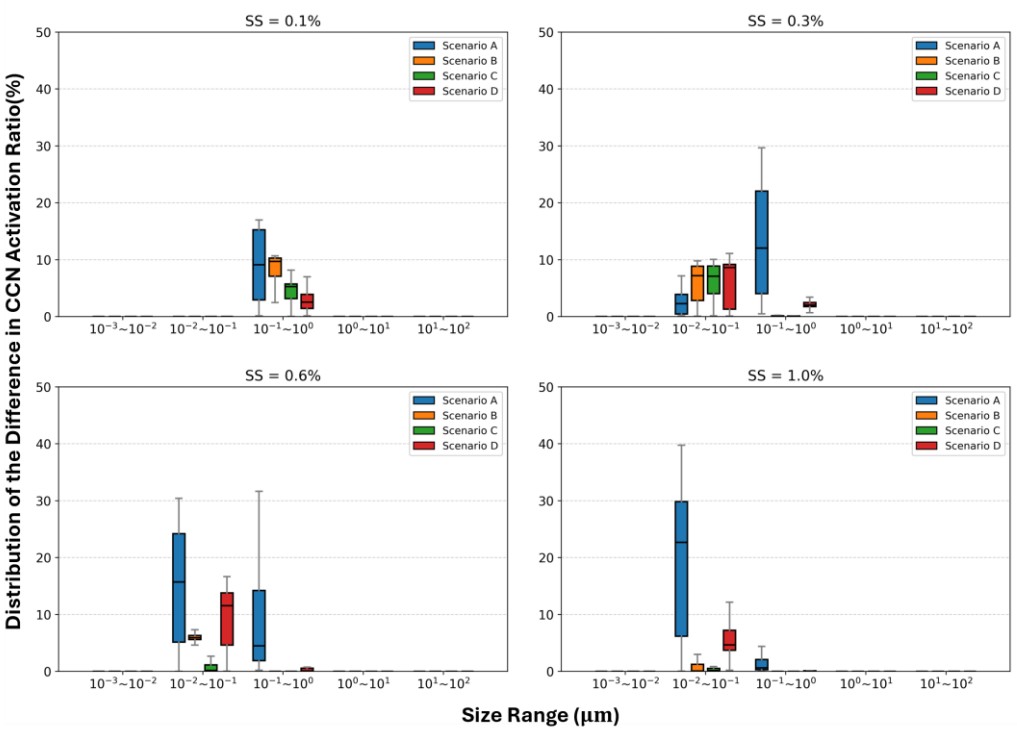


**Figure 6. The difference in CCN activation between composition-averaged and particle-resolved approaches within 5 logarithmic size bins ($10^{-3} \sim 10^2$ μm) at various environmental supersaturations for four scenarios. Boxplots show 25th–75th percentiles (with median line); data comes from ten-minute intervals over the simulation.**

Furthermore, the aerosol population is divided into 5 logarithmically spaced size bins (named Group 1 to Group 5 and ranging
from $10^{-3} \sim 10^2$ μm) to provide a more detailed analysis of the CCN prediction error from a size-resolved perspective, as shown in Fig. 6. Here, the composition-averaged method was performed separately within each size bin. The results aggregate data from the entire 24 hours at 10-minute intervals. They are also analyzed under four supersaturations, corresponding to Fig. 5. The boxplots represent the interquartile range ($25^{th} \sim 75^{th}$) and horizontal lines indicate medians. It can be seen that, in the smallest size bin (Group 1) and the larger size bins (Groups 4 and 5), although the hygroscopicity parameters calculated by the
two methods differ, particle activation is primarily determined by size rather than hygroscopicity. Therefore, under varying supersaturation levels, the sources of error are mainly concentrated in Group 2 ($10^{-2} \sim 10^{-1}$ μm) and Group 3 ($10^{-1} \sim 1$ μm),

which also account for the highest particle number concentrations. Compared to Group 2, the originally inactivated aerosols in the larger size bin (Group 3) are more likely to transition to an activated state as supersaturation increases. Consequently, the distribution of the CCN activation error caused by the composition-averaged method in Group 3 decreases with increasing

supersaturation. This underscores the importance of accurately determining the specific composition of particles to obtain more precise results. The deviation from the assumption of internal mixing emphasizes the importance of employing accurate measurement techniques or particle-resolved models like PartMC-MOSAIC, which can capture the real mixing state of aerosol particles. Such techniques can provide a more comprehensive understanding of the impact of the size distribution, chemical composition, and mixing state of the aerosols, thereby improving the accuracy of hygroscopicity calculations and related CCN

predictions.

**5 Conclusion**

Aerosol-cloud interactions remain a source of uncertainty in climate research. The current difficulty of cloud-altitude aerosol observations limits the investigation of the aerosol activation process at cloud-forming altitudes. The development of particle-

resolving simulation tools, such as the PartMC-MOSAIC model, facilitates the quantitative analysis of the hygroscopic behavior and cloud-forming potential of the aerosol populations. However, the infrequent consideration of the meteorological effects of transporting the air parcels to high altitudes could also introduce some errors or misunderstandings. To bridge the gap, a combination of cloud and particle-resolving models is utilized to investigate the evolution and activation properties of aerosols under idealized scenarios.

In this study, CM1 was applied to capture movement characteristics of air parcels under a typical shallow cumulus convection condition. Four distinct scenarios were devised based on the CM1 experiment. The parcel in the basic scenario maintains stability near the surface, while the others ascend to cloud-forming altitudes at varying intervals. Critical parameters extracted from CM1 simulations, including temperature, pressure, and parcel ascent timing, were subsequently fed into the PartMC-MOSAIC model. The observational and reanalysis data in the coastal area were also incorporated for initialization. Compared

to traditional frameworks, this PartMC-based approach offers a more comprehensive analysis of the specific chemical composition, hygroscopicity distribution, critical supersaturation distribution, and mixing state indices.

Following the ascent process into the free troposphere, aerosol-containing parcels cease to receive fresh emissions predominantly comprising hydrophobic substances and small particles. Rapidly ascending parcels may undergo significant environmental changes, such as a sharp temperature drop, impacting aerosol aging processes involving condensation and gas-

to-particle conversion. In this study, the ammonium nitrate variation is noteworthy owing to its high sensitivity to the temperature. The accelerated formation of these hygroscopic species enhances aerosol aging, elevates the median hygroscopicity parameter $\kappa$ of the aerosol population, and thereby impacts the particle activation properties. Compared to

aerosol populations near the surface, those at elevated altitudes mostly display a higher CCN activation ratio. The absolute ratio difference can reach up to 66% at a supersaturation of 1%, emphasizing the influence of vertical transport on aerosol activation potential.

Moreover, scenarios with different parcel ascent timings reveal marked variations in hygroscopicity and particle size distributions, primarily due to various accumulations of aerosol precursors. These precursors, such as nitric acid gas, ammonia, and ozone, are influenced by emission factors and photochemical reactions. In Scenario B, the parcel ascends before emissions and sunlight can take an important role, and the population shows the minimal median and overall $\kappa$ distribution among all the ascent scenarios. Hence, the cloud-forming potential is influenced by the ascent timing: the CCN activation in Scenario B could be as much as 15% lower than in the ground at a low supersaturation level (0.1%).

The combined consideration of hygroscopicity and size distribution is essential when estimating aerosol activation properties since some hygroscopic species may tend to grow on smaller particles. The PartMC model could capture detailed particle-based chemical composition data, which may reduce CCN prediction inaccuracies associated with simplified mixing-state assumptions. In our study, the internal mixing-state assumption results in a CCN activation ratio error from 7% to 30% compared with the particle-resolved results as supersaturation ranges from 0.1% to 1%.

In summary, the integration of particle-resolving models with dynamic meteorological simulations could mitigate the limitations of size-resolved methods and compensate for the deficiencies in cloud-altitude observations. The differences in aerosol properties between cloud-altitude and ground-level parcels highlight the significance of vertical aerosol observations and numerical modeling. Additionally, the mixing state index $\chi$ can be incorporated into future climate models to better constrain aerosol-cloud interactions. Shen et al. (2024) developed a machine learning model coupled to the Community Atmosphere Model Version 6 (CAM6) that can online correct biases in the mixing states. The $\chi$-guided adjustments provide a more accurate treatment of BC-related processes, including mixing state and coating status. In the future, $\chi$ has the potential to dynamically adjust other critical processes, offering a pathway to reduce microphysical uncertainties without requiring explicit microscale resolution.

**Code/Data availability**

The models used for the simulations, PartMC-MOSAIC and CM1, are open-source and can be obtained at https://github.com/compdyn/partmc and https://www2.mmm.ucar.edu/people/bryan/cm1/. The input parameters for the CM1 simulation can be downloaded from https://doi.org/10.5281/zenodo.15013770. The aerosol and gas data used in this study are freely available from papers listed in the Methodology and Supplementary section. The reanalysis dataset MERRA-2 is available at https://gmao.gsfc.nasa.gov/reanalysis/MERRA-2/. Other data is available after request.

## Author Contribution

XS and GY conceptualized and designed the study. GY collected and processed the data and YW assisted in data processing. GY performed simulations, analyzed data, wrote the original draft and edited the manuscript. ZW reviewed and contributed to the scientific discussions. XS supervised, reviewed, and edited the manuscript.

## Competing Interests

The authors declare that they have no conflict of interest.

## Acknowledgment

The work described in this paper was substantially supported by a grant from the Research Grants Council (RGC) of the Hong Kong Special Administrative Region, China (Project Reference: AoE/P-601/23-N). Additionally, XS by RGC grant HKUST-16307323. The authors thank HKUST Fok Ying Tung Research Institute and National Supercomputing Center in Guangzhou Nansha sub-center for providing high-performance computational resources.

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
