# Peer review of "Exploring the Aerosol Activation Properties in Coastal Shallow Convection Using Cloud and Particle-resolving Models"

_EGUsphere, 2024_

## Referee Comment (RC2)

Exploring the Aerosol Activation Properties in a Coastal Area Using Cloud and Particle-resolving Models

Yu et al.,

Summary

Yu and coauthors applied cloud parcel model and particle-resolved aerosol model to investigate under shallow convection conditions how aerosol evolution affect cloud-formation properties of the aerosol populations. They found that significant different between aerosols in boundary layer and high altitude in terms of CCN properties, also discrepancies in CCN activation ratio due to internal mixing assumption and that discrepancy increases with environmental supersaturation.

This study enhances our understanding of cloud formation properties of aerosol particles undergoing aging process under actual meteorological conditions when the air parcels leave the boundary layer. I would suggest the editor to consider accepting the manuscript for publication after emphasizing the novelty of this study and addressing the following questions or comments.

General and major comments

1. The unique contribution of this study is to incorporate shallow cumulus convention information to aerosol (particle-resolved) modeling in order to simulate the aerosol aging process (under shallow cumulus convection condition) and the associated CCN properties.

However, a lot of details about the methodology is missing. This part is important in that it is closely related to the conclusion of this study and provides unique contribution apart from existing studies of aerosol mixing state and CCN properties. Besides, lack of methodological details weakened the reproducibility and quality of this work, however this could be improved by providing more simulations details.

1.1 How is the Cloud Model 1 (CM1) configured to drive large-eddy simulations of ideal shallow cumulus convection? What are the specific model input and related parameters?

1.2 What do you mean by 'ideal shallow cumulus convection conditions'? Are these conditions based on any previous studies, any citations?

1.3 The authors tracked 484 parcels from the large eddy simulations, how to devise the four scenarios based on the 484 parcels? I don't see the technical details here in the manuscript. How do we know these 4 scenarios are representative? Besides, a table show the model input of these 4 scenarios is recommended for convenience.

1.4 Line 121-122, the authors extracted the temperature, pressure, kinetic diffusion coefficients for scenarios setup. Are these parameters the environmental input parameters to PartMC-MOSAIC simulations? How are relative humidity (or any other humidity measures) input to PartMC-MOSAIC?

1.5 Line 127-130 listed the four scenarios. What is meant by 'high altitude'?

1.6 Besides, it is not clear about the time dimension of the two set of model simulations, the CM1 and PartMC-MOSAIC. How long does the CM1 simulation last? 6 hours or 9 hours,10 hours or longer? It says PartMC-MOSAIC run for 30 hours, however, some features are shown e.g. concentration of Cl and nitrate increase drastically from the $2^{nd}$, $6^{th}$ and $10^{th}$ hour for scenarios B, C, and D respectively in Figure 1 (left panel). I wonder what is the relationship between the two 'time axes' of CM1 and PartMC-MOSAIC.

1.6 Also, it is not clear what is the relationship between the CM1 model output and PartMC-MOSAIC model input.

1.7 How do you deal with exchange of gases and aerosol particles (dilution) between the Lagrangian parcel of PartMC-MOSAIC and the surrounding environment? Is there any model output from CM1 that can guide PartMC-MOSAIC simulations in this regard?

1.8 Line 134, it says that horizontal eddy diffusivity at the high altitude was about one-fifth of the value near the surface. Is there any previous study supporting this assumption?

1.9 Line 140 and Table 1, how did authors use MERRA-2 reanalysis to derive concentration of aerosol at ground level and at high-altitude? A table showing the aerosol properties derived from MERRA-2 would be helpful, for example, this study is about coastal area, what is the concentration of sea salt aerosol and organic? How high is the 'high altitude'? Since authors also mentioned that vertical variability of aerosol is important, this vertical information is required to be clearly described.

2. About the quantification of the aerosol mixing state impact on cloud droplet formation property, presented in section 4 and figure 5, there is no details about how the composition averaging was performed. This is important because the conclusion about impact of mixing state and the error on CCN properties rely on this calculation.

3. Besides, there are already many studies about the relationship between aerosol mixing state and CCN properties of aerosol (the authored also cited some of these studies). What is the uniqueness and novelty of this model study? The authors emphasized their study incorporated shallow cumulus convention model, which is great. However, the authors are expected to explain the importance of such advancement and how does it compare to existing observational studies (even though there is no modeling study of similar kind).

4. Line 264, why scenarios B and C differ in aging process? More explanation is expected.

5. Line 286, why some hydrophilic species tend to form on smaller particles? What are those species? More explanation is expected.

6. Line 360, why at high supersaturation, CCN activation error becomes larger for some scenarios? More explanation is expected.

7. In figure 4b, why is there a peak for scenarios B, C and D at $2^{nd}$ and $6^{th}$ hour?

Specific comments

Apart from the general comments above, there are some specific questions or items to be clarified throughout the manuscript.

1. Section 4, (p.14) There is no details about the calculation of 'chi', the mixing state index. In the figure 4a, the D_alpha and D_gamma lie between 1.4 to 2.0, why is the range so small? There are so many chemical species listed in lines 160-164.

2. What is the definition of error in Figure 5? Does positive sign mean overestimation or underestimation of CCN by composition averaging? How about negative sign?

---

## Author Response (AR1)

**Response to referee comments**

Journal: Atmospheric Chemistry and Physics

Manuscript ID: Preprint egusphere-2024-3581

Title: Exploring the Aerosol Activation Properties in a Coastal Area Using Cloud and Particle-resolving Models

Authors: Ge Yu, Yueya Wang, Zhe Wang, Xiaoming Shi

We thank the reviewers for their constructive comments and suggestions, following which we revised the manuscript and improved its contents, structure, and quality. Below are our responses to the reviewers' comments. The text in black is the original comments from reviewers, and our **responses** are the **text in blue**. Some text from the revised manuscript is quoted in this response letter for the reviewers' convenience, and the *quoted text* is in *italic font*. The **line numbers** mentioned below are those in the revised PDF file, where the modification can be identified.

**Referee 1:**

This study uses the Cloud Model 1 (CM1) and the PartMC-MOSAIC model to simulate air parcel movement and aerosol evolution under shallow convection in coastal regions. The study concludes that transitioning from the boundary layer to the free atmosphere significantly affects aerosol properties, particularly through the rapid formation of ammonium nitrate, which increases the aerosols' cloud-forming potential. Additionally, the study highlights the importance of detailed aerosol mixing state representation for calculating CCN activity.

The premise of the paper, namely investigating aerosol aging processes as a function of altitude in the atmosphere, is interesting, as are the combination of tools (a cloud resolving model and a particle-resolved aerosol model). However, I have concerns about the setup of the simulations, about the novelty of the study, and about the generalization of the results. These need to be addressed before the paper can be considered for publication in ACP.

**Detailed comments are as follows:**

1. Simulation setup: There seems to be a mismatch between the PartMC-MOSAIC box model simulations and the way these are combined with the CM1 simulations. The authors use the scenario from Riemer et al. (2009), which was designed to represent the well-mixed boundary layer in a polluted urban region during the day (emissions are added to the simulation), and the residual layer during the night (no emissions added). The CM1 simulations on the other hand provide temperature, pressure, etc., of a particular parcel. If the CM1 simulations are used to drive the PartMC simulations, the authors need to think carefully about how emissions are added to the parcel. Specifically, once the parcel leaves the surface, no emissions should be added (i.e., even before it reaches the free troposphere), but mixing with the environment should potentially be considered.

Thank you for your detailed suggestions. We may not explain clearly in the initial methodology section. The utilization of Riemer's scenario primarily involves the distribution data of emission aerosols and gases. However, the emission temporal profiles have been adjusted based on our different parcels. Specifically, the parcel at the surface will continuously be affected by emitted substances, whereas parcels ascending from the surface have their emission sources closed upon departure (as you mentioned, no emissions), and this ascent process is also very rapid, completed within 10 minutes in PartMC.

Regarding the second point on the mixing with environmental substances, your considerations are very thorough. We indeed took note of this dilution process. For ascending parcels, we have adjusted the horizontal diffusion coefficient in the PartMC model (based on results from CM1) and set different background aerosol and gas concentrations at high altitudes. The conversion of gas and aerosol substances at high altitudes will be explained specifically under the ninth comment.

We have modified the method section to make the description clear, relevant part is pasted below (lines 125~128, 141~146):

*"After updraft, the temperature rapidly dropped to 17 ℃ (290 K) and the pressure became 880 hPa, corresponding to approximately 1.2 km altitude. As the parcel ascended and remained at a high altitude, the emission from ground sources ceased to enter these parcels."*

*"Subsequently, we converted the data and generated the high-altitude background aerosol used in the simulation. The substances in the ascent parcels would dilute with high-altitude background aerosols and gases. Detailed information on the number concentration and size distribution of the aerosols in different modes can be found in Table 2. The initial aerosol conditions in the parcels contain three modes that are the same as the Ground part because all the parcels stay near the surface during the initialization period. The background concentrations for ground-level and high-altitude gases are listed in Supplementary Information S5."*

2. Novelty: Ching et al. (2017, ACP, 17, 7445-7458) have investigated mixing state impacts on CCN activity in great detail. Granted, they did not use CM1 to construct trajectories to drive PartMC. However, at the end of the day, what do we learn from your study that is new and unique compared to Ching et al. (2017)?

Thanks for your question. This study was indeed inspired by the work of Ching et al. (2017, ACP, 17, 7445-7458), which provided a quantitative explanation of the relationship between mixing state and errors in CCN concentrations. Their work offered valuable insights into the analysis of aerosol mixing states and their impact on CCN activity. However, our study introduces several unique and innovative aspects.

Currently, numerous large-scale meteorological simulations employ relatively simple aerosol parameterization methods when considering the contribution of CCN to cloud microphysical processes, such as directly prescribing aerosol activation rates. These simplifications may introduce significant errors in the prediction of cloud behavior and associated atmospheric processes. This study integrates the meteorological model CM1 with the aerosol evolution model PartMC for the first time. By leveraging CM1, we derived the motion characteristics of air parcels in a coastal area and analyzed the evolution of aerosols from a particle-resolved perspective during parcel transport. This combined approach has built a framework on the aerosol simulations considering the meteorological condition. Although this study focuses on a smaller scale compared to typical meteorological simulations, it lays an important foundation for improving CCN parameterization schemes in large-scale models.

In addition, although PartMC has been extensively applied in previous studies, these studies primarily focused on parcels within the boundary layer. In contrast, this work includes the ascent of parcels into the free troposphere due to meteorological influences, where cloud formation occurs. During this ascent, the environmental conditions of the parcels undergo significant changes (e.g., temperature, pressure, and surrounding gas), which in turn affect aerosol evolution and CCN activation. This aspect is one of the main contributions of our work, as it evaluates aerosol hygroscopicity and cloud-forming potential under realistic cloud-forming conditions.

Besides, this paper reveals that not only the updraft of parcels but also the timing of their ascent will influence the aerosol evolution. The factors affecting aerosol aging processes, including chemical composition, particle size, mixing state and their impact on CCN activation are quantified through systematic analysis of four typical scenarios. The advantages of the particle-resolved model over the composition-averaged method are compared and identified.

We accordingly revised the manuscript and now the novelty mentioned in the introduction section is as follows:

*"Although the PartMC-MOSAIC has been extensively applied in previous studies, these studies primarily focused on parcels within the boundary layer, this may limit the accurate representation of environmental conditions when studying the cloud-forming ability of aerosols. Compared to the work mentioned above,*

*Curtis et al. (2017) noted that PartMC, as a zero-dimensional box model, lacks spatial information and has not been integrated into meteorological studies. To address this, they pioneered the coupling of WRF with PartMC, resolving aerosol composition on a per-particle level and integrating aerosol chemistry with meteorology. This represents one of the few studies to combine meteorological modeling with PartMC, which reveals the potential of PartMC in studying aerosol vertical transport. Building on this foundation, our study further explores the application of meteorological simulations coupled with PartMC, advancing the understanding of this approach in aerosol research.*

*Currently, large-scale meteorological simulations generally employ relatively simple aerosol parameterization methods when considering the contribution of CCN to cloud microphysical processes (Hazra et al., 2020; Morrison and Milbrandt, 2011; Thompson and Eidhammer, 2014), such as directly prescribing aerosol activation rates. These simplifications may introduce errors in the prediction of cloud behavior and associated atmospheric processes. To this end, this study integrates the meteorological Cloud Model 1 (CM1) (Bryan and Fritsch, 2002) with the aerosol evolution model PartMC for the first time."*

3.    Figure S1 shows cloud water/ice mixing ratios. Is it cloud water or ice? Are the aerosols interacting with the existing clouds in any way? I assume they don't. What does this mean for the realism of the aerosol simulations? This deserves some discussion. I also noticed that the cloud water mixing ratios are quite small. Could the authors comment on this?

The plot you mentioned in Figure S1 displays the combination mixing ratio of cloud water and ice. Based on this question, we clarified it in the caption for Figure S1 in the new supplementary (lines 5 ~ 6). Figure S1 and the modified caption are pasted below:

[Figure]

**Figure S1. The temporal changes of the potential temperature, water vapor mixing ratio, and cloud water and ice mixing ratio with the height in the preliminary CM1 experiment. The results shown in the figures are the average values on the horizontal plane.**

Yes, as you mentioned, our aerosol simulation does not involve interactions with clouds. Thank you for your suggestion, and we have added some explanations in the new supplementary section (lines 16 ~ 18):

*"The presentation of this plot is to illustrate the potential cloud formation heights under the typical*

*shallow cumulus convection conditions. This also highlights the significance of scenario design in our study. We aim to obtain air parcels at corresponding heights where clouds can form to assess the aerosol evolution processes and the particles' cloud-forming potential."*

Regarding the issue of "the cloud water mixing ratios are quite small," this is because the values presented are averaged mixing ratios on the horizontal plane (x-y plane). We have added some descriptions in the caption (line 6) and we appreciate your attention to detail. Since clouds only form in the middle regions of a plane at a specific height, the surrounding blank regions lower the average cloud-water mixing ratio at that height. Therefore, the results presented show relatively small values.

4. Since the paper is about CCN properties in a coastal area, I would assume that there should be sea spray aerosol, however the paper mentions that sodium chloride concentrations are very small. Please explain.

Thank you for your meticulous review. The aerosol data we employed were derived from observations conducted at the station in Tsuen Wan, Hong Kong (Wang et al., 2017). The measurement site is not directly adjacent to the coast and larger sea salt particles typically do not disperse over large distances from the coastline. Consequently, the observed data and our initial inputs encompass relatively lower levels of sodium chloride than in marine environments.

The initial aerosol data we utilized, from the summer averages, represent the background aerosols in coastal urban areas such as Hong Kong. These initial conditions, also being background aerosols, impact the final simulated concentrations of NaCl.

Moreover, our objective is to establish a framework through idealized simulations. The selected setting for these simulations is a coastal area, primarily due to the tendency for clouds to form at relatively low altitudes. Shallow cumulus convection conditions are common in these areas. The framework in this work will enable future researchers to input aerosol and gas emission profiles tailored to their specific areas of interest (e.g., adjustments for scenarios with high sea salt content), facilitating the achievement of results that better align with new research needs.

5. The nitrate concentrations are very high. Please provide some context where (in the world) such high nitrate concentrations could be encountered. How generalizable are these results? It would be helpful to have simulations with lower nitrate levels.

Thank you for your question. We have sourced the background nitrate data from the Hong Kong observation station (Wang et al., 2017). Through the confirmation, we found that the measurements reported by Wang and Yu (2017) were conducted during 2013~2014. The observational data is approximately one decade old, during which nitrate pollution levels were substantially higher than present conditions. In our study, the total background nitrate concentration (summing condensation, droplet, and

coarse modes) was set at 1.20 μg/m³ for summer conditions.

During the observation period, nitrate pollution in China was particularly severe. Several studies from that time report nitrate concentrations comparable to the values used in our study. For instance, Figure 5 in Elser et al. (2016) presents reference diurnal trends of PM2.5 components (excluding extreme haze events) observed during 2013–2014, showing nitrate concentrations reaching approximately 10 μg/m³ in Xi'an and Beijing.

[Figure]

**Figure 5.** Diurnal trends, size distributions, mean concentrations (NR-PM$_{2.5}$ plus eBC mass) and relative contributions of the AMS species and eBC for the four periods. Note: size distributions only available for AMS species.

Additionally, Figure 1 in Zhai et al. (2021) summarizes annual mean nitrate concentrations in PM2.5 across China during 2013–2015, with some coastal regions exceeding 10 μg/m³.

[Figure]

**Fig. 1 | PM$_{2.5}$ nitrate concentrations in China and comparisons between observations and GEOS-Chem model results. a,b,** Surface air PM$_{2.5}$ nitrate concentrations from two nationwide datasets (circles) and GEOS-Chem (background) for 2013 (**a**, annual mean) and 2015 (**b**, summer and winter mean). The colour bar shows PM$_{2.5}$ nitrate concentration in µg m$^{-3}$. **c,d,** Scatterplots of observed and modelled winter (filled circles) and summer (open circles) monthly mean (seasonal mean for the 2015 dataset) nitrate at individual sites. Also shown in **c** and **d** are the 1:1 lines, the wintertime correlation coefficients (*r*) between model and observations, and the corresponding reduced-major-axis regressions and slopes (±95% confidence interval). The 2013 dataset is from the Campaign on Atmospheric Aerosol Research network of China (CARE-China), with nitrate measured by ion chromatography[36,48]. The 2015 dataset is from ref. [37], and only includes sites that have both winter and summer observations. The dashed rectangle in **a** delineates the North China Plain region as defined in this paper (113.75°–118.75° E, 35°–41° N).

We adopted the Hong Kong observational data for parcel initialization and background conditions because it provides comprehensive aerosol speciation that meets our model's input requirements. The subsequent nitrate increase in basic Scenario A relates to urban background ammonia and nitric acid concentrations specified in emission inventories. Photochemical processes drive ozone production and NOx oxidation, leading to rapid daytime nitrate accumulation with limited dispersion. Actual nitrate removal mechanisms (including dispersion and deposition) are strongly influenced by meteorological conditions (Zhai et al., 2021), which are not currently represented in our 0D PartMC-MOSAIC model.

The elevated nitrate concentrations observed in parcels after ascent result from complex interactions between rapid environmental changes and background substances. Nitrate aerosols exhibit strong temperature sensitivity. Gas-particle partitioning occurs within parcels experiencing significant temperature fluctuations. These results represent nitrate levels within localized air parcels rather than ambient atmospheric concentrations at equivalent altitudes.

The current simulations aim to establish a framework, which is applicable to various scenarios and background substance data, including lower nitrate concentrations. We have carefully considered the reviewer's concerns and added explanations regarding our data selection in the revised manuscript. For reviewers' reference, we have added the following clarification in lines 141~142:

*"The nitrate levels observed in Hong Kong (used in this study) may be higher than other coastal cities*

*due to historical pollution patterns from a decade ago, but the methodology can be adapted to incorporate different observational data from other regions."*

References for nitrate concentration:

Elser, M., Huang, R.-J., Wolf, R., Slowik, J. G., Wang, Q., Canonaco, F., Li, G., Bozzetti, C., Daellenbach, K. R., Huang, Y., Zhang, R., Li, Z., Cao, J., Baltensperger, U., El-Haddad, I., & Prévôt, A. S. H. (2016). New insights into $PM_{2.5}$ chemical composition and sources in two major cities in China during extreme haze events using aerosol mass spectrometry. *Atmospheric Chemistry and Physics*, *16*(5), 3207–3225. https://doi.org/10.5194/acp-16-3207-2016

Zhai, S., Jacob, D. J., Wang, X., Liu, Z., Wen, T., Shah, V., Li, K., Moch, J. M., Bates, K. H., Song, S., Shen, L., Zhang, Y., Luo, G., Yu, F., Sun, Y., Wang, L., Qi, M., Tao, J., Gui, K., … Liao, H. (2021). Control of particulate nitrate air pollution in China. *Nature Geoscience*, *14*(6), 389–395. https://doi.org/10.1038/s41561-021-00726-z

6.    Please add more information about the individual trajectories. What are the temperatures (what range is covered between the different trajectories), what is the RH (range?), and what are the gas phase concentrations?

Thank you for your suggestion. We have conducted extraction and analysis of key variables (air temperature and relative humidity) associated with CM1-tracked parcels, with the compiled data systematically organized in Table R1.

**Table R1. The range of temperature and RH in the parcels of CM1 simulation.**

| Conditions | Temperature range (K) | RH range (%) |
|---|---|---|
| Ascent parcels | 287 - 300 | 57 - 100 |
| All the tracked parcels | 287 - 300 | 57 - 100 |

Utilizing CM1 output, we implement scenario-specific temperature variations in PartMC simulations. The data is extracted from representative parcels. In different trajectories, the parcel temperature at ground level is set as 299 K and at elevated altitude as 290 K. Due to the characteristics of the PartMC model, we first set the initial relative humidity to 60%, which increases to 92% after the parcel ascends based on temperature changes. We have revised the relevant descriptions in the method section. The revised content is now in lines 123~128:

*"For the parcels that stay at the surface, the background temperature was set to 26 ℃ (299 K), and the pressure was set to the standard atmospheric pressure. After updraft, the temperature rapidly dropped to 17 ℃ (290 K) and the pressure became 880 hPa, corresponding to approximately 1.2 km altitude. As the*

*parcel ascended and remained at a high altitude, the emission from ground sources ceased to enter these parcels. To align with the humidity conditions of ascending parcels in CM1, the initial relative humidity was set to 60%. Environmental parameter configurations for different scenarios are listed in Table 1."*

**Table 1. Fundamental settings for 4 different scenarios.**

| Scenario | Ascent Timing (hr) | Temperature before Ascent (K) | Temperature after Ascent (K) | Pressure before Ascent (hPa) | Pressure after Ascent (hPa) | Initial Relative Humidity (%) |
|---|---|---|---|---|---|---|
| A | - | 299 | - | 1000 | - | 60 |
| B | 2 | 299 | 290 | 1000 | 880 | 60 |
| C | 6 | 299 | 290 | 1000 | 880 | 60 |
| D | 10 | 299 | 290 | 1000 | 880 | 60 |

Initial gas concentrations were consistent with surface background levels and evolved dynamically throughout temporal development. The simulation ranges for gas concentrations are tabulated below and have been appended to Supplementary Section S5. The supplemental additions include:

**Table S1. The Background Conditions and Emissions for Gases**

| Gas Species | Symbol | Ground Fraction (ppb) | Emissions (nmol m² s⁻¹) | High-altitude Fraction (ppb) |
|---|---|---|---|---|
| Nitric oxide | NO | 0.1 | 31.8 | 0.091 |
| Nitrogen dioxide | $NO_2$ | 1.0 | 1.67 | 0.91 |
| Nitric acid | $HNO_3$ | 1.0 | | 0.91 |
| Ozone | $O_3$ | 50.0 | | 45.5 |
| Hydrogen peroxide | $H_2O_2$ | 1.1 | | 1.001 |
| Carbon monoxide | CO | 21.0 | 291.3 | 19.11 |
| Sulfur dioxide | $SO_2$ | 0.8 | 2.51 | 0.728 |
| Ammonia | $NH_3$ | 0.5 | 6.11 | 0.455 |
| Hydrogen chloride | HCl | 0.7 | | 0.637 |
| Methane | $CH_4$ | 2200.0 | | 2002 |
| Ethane | $C_2H_6$ | 1.0 | | 0.91 |
| Formaldehyde | HCHO | 1.2 | 1.68 | 1.092 |

| | | | | |
|---|---|---|---|---|
| Methanol | CH$_3$OH | 0.12 | 0.28 | 0.1092 |
| Methyl hydroperoxide | CH$_3$OOH | 0.5 | | 0.455 |
| Acetaldehyde | ALD$_2$ | 1.0 | 0.68 | 0.91 |
| Paraffin carbon | PAR | 2.0 | 96.0 | 1.82 |
| Acetone | AONE | 1.0 | 1.23 | 0.91 |
| Ethene | ETH | 0.2 | 7.2 | 0.182 |
| Terminal olefin carbons | OLET | 0.023 | 2.42 | 0.02093 |
| Internal olefin carbons | OLEI | 0.00031 | 2.42 | 0.0002821 |
| Toluene | TOL | 0.1 | 4.04 | 0.091 |
| Xylene | XYL | 0.1 | 2.41 | 0.091 |
| Lumped organic nitrate | ONIT | 0.1 | | 0.091 |
| Peroxyacetyl nitrate | PAN | 0.8 | | 0.728 |
| Higher organic acid | RCOOH | 0.2 | | 0.182 |
| Higher organic peroxide | ROOH | 0.025 | | 0.02275 |
| Isoprene | ISOP | 0.5 | 0.23 | 0.455 |
| Alcohols | ANOL | | 3.45 | 0.091 |

*"The gaseous species inputs across all simulation scenarios are systematically cataloged in Table S1. These chemical species align with those documented in Zaveri and Peter (1999), while surface background concentrations and emission parameters were adopted from Riemer et al. (2009). For elevated atmospheric conditions, we implemented the ideal gas Clapeyron equation with the following boundary conditions:*

*High altitude: 880 hPa, 290 K*

*Ground level: 1000 hPa, 299 K*

*These conditions enabled the derivation of trace gas concentrations at the high altitude of the ascent parcels. Emission rates presented in the table represent temporal averages over emission periods, with actual rates subject to modulation by mixing height variations* (Riemer et al., 2009a)."

7.    The first two paragraphs of the introduction are too generic. I recommend that the authors get to the point more quickly – how mixing state impacts CCN properties, what is known about this, and what the novel contribution of this paper is. Additionally, the references for some statements are not well chosen, e.g., Mishra et al., (2023), Curtis et al. (2017), Lolli et al. (2023). Please carefully check to make sure that each reference really supports the statement made.

Thank you for your highly practical suggestions. We have restructured the first two paragraphs by merging them into a single paragraph to enable a quicker transition into the main topic. In the second paragraph, we directly introduced the theme of the mixing state's impact on CCN. The second paragraph of the revised introduction is as follows:

*"The mixing state indicates the chemical composition distribution among aerosol particles, the treatment of which represents a crucial aspect in investigating aerosol aging processes and CCN activation properties. There are two ideal extremes of the mixing state: internally mixed and externally mixed (Winkler, 1973) commonly applied as simplified assumptions. The former assumes that each particle shares the same chemical composition, while the latter considers that each individual particle contains distinct species. In aerosol research, Riemer introduced the use of the mixing state index to quantitatively assess the degree of mixing state (Riemer and West, 2013). Subsequently, Ching et al. provides a detailed explanation of the impact of mixing state on CCN properties. They proved that simplifying assumptions about mixing state will cause inevitable errors due to the loss of particle size information (Ching et al., 2017). However, such simplifications regarding the mixing state remain inevitable, especially in experimental research and large-scale simulations."*

Additionally, we have double-checked and ensured the citation of each reference to better support the statements. We greatly appreciate your attention to this issue. The work by Curtis et al. has now been cited in the section discussing PartMC-related studies (lines 74~78).

8.    The authors refer with "kappa" somewhat loosely to both a population-level "bulk" kappa and to a per-particle kappa. It would be helpful if the authors could clearly explain early on (using appropriate notation) that aerosol particles should be described by a distribution of kappa values, just like they are described by a distribution of sizes. From there we can average within size ranges or within the whole population to arrive at an average kappa for the population

Your suggestion helps to clarify our article. We have incorporated some descriptions into the original text of the methodology section (lines 181~185):

*"For the particle $i$ containing various components, $\kappa_i$ is the volume-weighted average of the $\kappa$ values of its constituent aerosol species. For the aerosol population, the median value $\kappa_m$ of all $\kappa_i$ values is calculated to serve as a characteristic parameter of the ensemble hygroscopicity, which is analyzed in Section 3. Table 3 lists specific $\kappa$ values assigned to different species simulated in this study (Clegg et al.,*

1998; Riemer et al., 2009b; Zaveri et al., 2010).″

Correspondingly, we have revised the Figure 2 caption and κ-related descriptions in the Results section. The manuscript now systematically employs $\kappa_i$ for per-particle hygroscopicity parameters, $\kappa_m$ for population-level median values, and $\kappa$ (without subscript) to denote the characteristic hygroscopicity of sub-populations. The revised caption for Figure 2 is pasted below:

[Figure]

**Figure 2. The median hygroscopicity parameter $\kappa_m$ changes over time and the number concentration distribution of the particles with different $\kappa_i$ values across particle size at 0, 6, 12, and 24 hours in (a) Scenario A, (b) Scenario B, (c) Scenario C, and (d) Scenario D. The blue solid line in the left plots represents the median $\kappa$ across the aerosol population, while the edges of the green shading denote the 25th and 75th percentile $\kappa_i$ values.**

9.  Line 142: How was the conversion to high-altitude background aerosol data done?

Thank you for your careful consideration of these details. The data conversion methodology applies ground-based measurements (Wang et al., 2017) and three-dimensional aerosol reanalysis data from MERRA-2 (the Modern-Era Retrospective Analysis for Research and Applications, Version 2).

The observation data contains total mass concentrations of various aerosol species along with their multimodal size distribution parameters. The employed MERRA-2 data (M2I3NVAER, DOI: 10.5067/LTVB4GPCOTK2) features instantaneous 3D fields with 3-hourly temporal resolution, vertically resolved into 72 atmospheric layers (layer 72 corresponds to the surface at 1000 hPa). The aerosol components include sulfate, black carbon, organic carbon, and sea salt, maintaining consistency with the observational species.

We extracted MERRA-2 aerosol data for the Hong Kong region (113.9°E-114.6°E, 22.2°N-22.5°N) using its native 0.625° longitude × 0.5° latitude grid. The data of the lowest 20 atmospheric layers (from 880 to 1000 hPa) in summer (June to August) was extracted and processed. Subsequent calculations determined the mass concentrations of different species at 880 hPa (the high-altitude background aerosol) through vertical proportionality relationships between surface (1000 hPa) and target altitude. To meet PartMC's requirement for aerosol inputs, we converted the data and generated the high-altitude background aerosol data used in simulations.

We have accordingly updated the methodology section (Section 2.1) in the revised manuscript to elaborate these technical procedures with enhanced clarity (lines 138~144):

*"We extracted the reanalysis aerosol data of Modern-Era Retrospective Analysis for Research and Applications, Version 2 (MERRA-2, DOI: 10.5067/LTVB4GPCOTK2) in the Hong Kong region. The data of the lowest atmospheric layers (from 880 to 1000 hPa) in summer was processed and the vertical proportionality relationships between the ground-level aerosol species and high-altitude species were estimated. Subsequently, we converted the data and generated the high-altitude background aerosol used in the simulation. The substances in the ascent parcels would dilute with high-altitude background aerosols and gases. Detailed information on the number concentration and size distribution of the aerosols in different modes can be found in Table 2."*

New added references:

Global Modeling and Assimilation Office (GMAO) (2015), inst3_3d_aer_Nv: MERRA-2 3d,3-Hourly, Instantaneous, Model-Level, Assimilation, Aerosol Mixing Ratio, version 5.12.4, Greenbelt, MD, USA: Goddard Space Flight Center Distributed Active Archive Center (GSFC DAAC), Accessed Enter User Data Access Date at doi: 10.5067/LTVB4GPCOTK2. Retrieved from https://gmao.gsfc.nasa.gov/reanalysis/MERRA-2/.

10. Figure 1: To help the reader, in the caption please repeat what cases A, B, C, D are.

Thank you for your suggestion. We have added brief descriptions for the four different cases in the caption of Figure 1. This can be found in lines 201~202 of the revised manuscript.

Additionally, we have modified the images to make the various substances more distinct and friendly to readers with color vision deficiencies. The new figure and caption are pasted below:

[Figure]

**Figure 1. Mass concentration variations over time of hydrophilic and carbonaceous components in Scenario A, B, C, and D. For Scenario A: the parcel remains near the ground; B, C and D: the parcel ascends after 2 hours, 6 hours and 10 hours respectively.**

11. Figure 2: How is kappa for the population derived from the kappa_i (i.e., weighted by the particle mass?)?

The representative kappa value $\kappa_m$ for the population is determined by calculating the median $\kappa_i$ value of all particles. As a result, the method relies on particle number counts rather than mass.

Due to the nature of the model, each output computational particle is associated with a number concentration. When calculating the median and the 25th and 75th percentiles, we also considered using the number concentration as a weighting factor.

12. How was the composition averaging done – over the whole population, or within certain size ranges, e.g., fine and coarse? A size-resolved treatment could be interesting because the error probably comes from a narrow size range (which also depends on supersaturation).

The composition averaging was done over the whole population. We statistically aggregated the chemical compositions of all particles, assuming each aerosol population exists as a totally internal mixture. We have added some explanations in lines 353~354:

*"Due to measurement limitations on aerosol components, the hygroscopicity of individual particles is often estimated using the average chemical composition of the total aerosol population, which is based*

*on the fully internal mixing assumption."*

We sincerely appreciate your valuable suggestions. The size-resolved perspective proves particularly insightful: neglecting mixing state effects may substantially influence particles within specific size regimes. We have implemented a logarithmic size binning (5 categories) approach and conducted corresponding parametric analysis, with results comprehensively detailed in Figure 6 of the revised manuscript. We have restructured Section 4 to enhance clarity and logical flow. The newly added content, which addresses these critical aspects, is presented below:

*"4.2 CCN calculation error analysis from size-resolved perspective*

[Figure]

**Figure 6. The difference in CCN activation between composition-averaged and particle-resolved approaches within 5 logarithmic size bins ($10^{-3}$ ~ $10^2$ μm) at various environmental supersaturations for four scenarios. Boxplots show 25th–75th percentiles (with median line); data comes from ten-minute intervals over the simulation.**

*Furthermore, the aerosol population is divided into 5 logarithmically spaced size bins (from 1 to $10^5$ nm) to provide a more detailed analysis of the CCN prediction error from size-resolved perspective, as shown in Fig. 6. Here, the composition-averaged method was performed separately within each size bin. The results aggregate data from the entire 24 hours at 10-minute intervals. They are also analyzed under four supersaturations, corresponding to Fig. 5. The boxplots represent the interquartile range (25th–75th) and horizontal lines indicate medians. It can be seen that, in the smallest size bin (Group 1) and the larger size bins (Groups 4 and 5), although the hygroscopicity parameters calculated by the two methods differ, particle activation is primarily determined by size rather than hygroscopicity. Therefore, under varying supersaturation levels, the sources of error are mainly concentrated in Group 2 (10~100 nm) and Group 3 (100~1000 nm), which also account for the highest particle number concentrations. Compared to Group*

*2, the originally inactivated aerosols in the larger size bin (Group 3) are more likely to transition to an activated state as supersaturation increases. Consequently, the distribution of the CCN activation error caused by the composition-averaged method in Group 3 decreases with increasing supersaturation. This underscores the importance of accurately determining the specific composition of particles to obtain more precise results. The deviation from the assumption of uniform mixing emphasizes the importance of employing accurate measurement techniques or particle-resolved models like PartMC-MOSAIC, which can capture the real mixing state of aerosol particles. Such techniques can provide a more comprehensive understanding of the impact of the size distribution, chemical composition, and mixing state of the aerosols, thereby improving the accuracy of hygroscopicity calculations and related predictions."*

13. Figure 5, y-axis, suggest to display error in percent rather than as a fraction.

According to your helpful suggestions, we have made modifications to Figure 5. Based on these changes, the discussion in section 4.1 is restructured as well. The new figures are pasted below:

[Figure]

**Figure 5. The difference in CCN activation between composition-averaging and particle-resolved results with the mixing state index $\chi$ at various environmental supersaturations for four scenarios. The scatters show the results at ten-minute intervals.**

**Referee 2:**

Yu and coauthors applied cloud parcel model and particle-resolved aerosol model to investigate under shallow convection conditions how aerosol evolution affect cloud-formation properties of the aerosol populations. They found that significant different between aerosols in boundary layer and high altitude in terms of CCN properties, also discrepancies in CCN activation ratio due to internal mixing assumption and that discrepancy increases with environmental supersaturation.

This study enhances our understanding of cloud formation properties of aerosol particles undergoing aging process under actual meteorological conditions when the air parcels leave the boundary layer. I would suggest the editor to consider accepting the manuscript for publication after emphasizing the novelty of this study and addressing the following questions or comments.

**General and major comments**

1.   The unique contribution of this study is to incorporate shallow cumulus convention information to aerosol (particle-resolved) modeling in order to simulate the aerosol aging process (under shallow cumulus convection condition) and the associated CCN properties. However, a lot of details about the methodology is missing. This part is important in that it is closely related to the conclusion of this study and provides unique contribution apart from existing studies of aerosol mixing state and CCN properties. Besides, lack of methodological details weakened the reproducibility and quality of this work, however this could be improved by providing more simulations details.

1.1  How is the Cloud Model 1 (CM1) configured to drive large-eddy simulations of ideal shallow cumulus convection? What are the specific model input and related parameters?

Thank you for the question. The CM1 model was configured with a horizontal grid spacing of 25 m and a vertical spacing of 25 m. The domain size was 6.3 km × 6.3 km × 3 km. Initial conditions included a well-mixed boundary layer with surface heating and a dry adiabatic lapse rate above the boundary layer to trigger shallow convection.

These settings align with idealized shallow cumulus studies (Bryan & Fritsch, 2002; Siebesma et al., 2003). Regarding the specific model inputs and parameters, we have uploaded the namelist file of CM1 we used for simulations on the Zenodo website (DOI: 10.5281/zenodo.15013770). We have accordingly revised the code/data availability section of the manuscript: *"The input parameters for the CM1 simulation can be downloaded from https://doi.org/10.5281/zenodo.15013770."*

1.2  What do you mean by 'ideal shallow cumulus convection conditions'? Are these conditions based on any previous studies, any citations?

'Ideal shallow cumulus conditions' refer to a setup where surface heating drives buoyant plumes that form non-precipitating cumulus clouds, which is normal in the coastal area.

Yes, these conditions are based on some previous classical LES studies (e.g., Siebesma et al., 2003) and avoid complexities like wind shear or moisture stratification. Thanks for your suggestion and we added the citation in Section 2.1, line 100.

New added references:

Siebesma, A. P., Bretherton, C. S., Brown, A., Chlond, A., Cuxart, J., Duynkerke, P. G., Jiang, H., Khairoutdinov, M., Lewellen, D., Moeng, C.-H., Sanchez, E., Stevens, B., and Stevens, D. E.: A Large Eddy Simulation Intercomparison Study of Shallow Cumulus Convection, 2003.

1.3   The authors tracked 484 parcels from the large eddy simulations, how to devise the four scenarios based on the 484 parcels? I don't see the technical details here in the manuscript. How do we know these 4 scenarios are representative? Besides, a table show the model input of these 4 scenarios is recommended for convenience

Thank you for your question. In the CM1 simulations, we tracked the evolution of parcel conditions (height, temperature, pressure, moisture, etc.). Statistical analysis revealed that approximately one-third of the parcels ascended to altitudes exceeding 1,000 meters during the hours of shallow cumulus convection. These parcels typically rose rapidly within 20 minutes, with ascent timing being highly stochastic throughout the weather event. The remaining air parcels persisted in the near-surface layer below 500 meters.

Among the four scenarios implemented, Scenario A was devised to remain near the surface throughout, exhibiting fundamentally distinct characteristics that provide representative baseline comparisons. Shallow convective events typically occur under inhomogeneous atmospheric thermal conditions, frequently observed in coastal areas with daytime solar radiation. Our PartMC simulations spanned from 06:00 to 06:00 the following day (detailed further under question 1.6). To align with the most probable timing of shallow convection, we scheduled parcel ascent around noon (12:00) ± several hours. Additionally, to analyze how aerosol population evolution is influenced by ground-level residence time (for emission exposure), we designed Scenarios B, C, and D with three distinct ascent timings. Equal time intervals were controlled to ensure representativeness and align with the stochastic ascent timing observed in CM1 simulations.

Some relevant results from the CM1 simulations are listed in Supplementary Material S1, with partial explanations provided as follows:

*"In this weather process, clouds and ice clouds gradually form between 500 m and 2000 m. Most of the tracked air parcels tend to move within 2000 m. As mentioned in the main text, approximately one-third of the air parcels ascend to altitudes exceeding 1000 meters, where cloud formation occurs. The ascent of air parcels is rapid, leading to significant environmental changes during this period."*

As suggested, we have modified the technical descriptions and compiled the input conditions

(temperature, pressure, relative humidity) for these 4 scenarios in the new Table 1. Modified tables and explanations in section 2.1 are also provided below:

*"Shallow cumulus convection is more likely to be triggered around noon when surface heat flux is strongest. The CM1 simulation captured several hours of this meteorological phenomenon, while the subsequent aerosol simulations using the PartMC model would span 24 hours (starting at 6 a.m.). Given the disparities between the surface level and cloud-forming altitudes, we first designed a baseline scenario where the parcels remain near the ground throughout the entire day. To align with the most probable timing for shallow convection, the parcel ascent events were scheduled around noon (12:00) with a temporal tolerance of 4 hours."*

*"Environmental parameter configurations for different scenarios are listed in Table 1."*

**Table 1. Fundamental settings for 4 different scenarios.**

| Scenario | Ascent Timing (hr) | Temperature before Ascent (K) | Temperature after Ascent (K) | Pressure before Ascent (hPa) | Pressure after Ascent (hPa) | Initial Relative Humidity (%) |
|---|---|---|---|---|---|---|
| A | - | 299 | - | 1000 | - | 60 |
| B | 2 | 299 | 290 | 1000 | 880 | 60 |
| C | 6 | 299 | 290 | 1000 | 880 | 60 |
| D | 10 | 299 | 290 | 1000 | 880 | 60 |

1.4 Line 121-122, the authors extracted the temperature, pressure, kinetic diffusion coefficients for scenarios setup. Are these parameters the environmental input parameters to PartMC-MOSAIC simulations? How are relative humidity (or any other humidity measures) input to PartMC-MOSAIC?

Yes, these parameters were directly input into PartMC-MOSAIC for the parcel simulation process. Relative humidity (RH) was derived from CM1's moisture fields. However, in the PartMC model, unlike environmental variables such as temperature and pressure (which directly follow prescribed temporal profiles), RH is input just with an initial value. Subsequent RH changes are calculated based on temperature under the assumption of constant total water content.

We statistically analyzed the relative humidity range of parcels in CM1 (57–100%). For ascending parcels, RH increased from ~60% (surface) to ~90% (cloud level). The figure below illustrates the temperature and relative humidity variations of a representative ascending parcel in CM1.

[Figure]

In PartMC, we set identical initial RH values of 60% for all scenarios, with RH rising to approximately 92% after ascent. Sensitivity tests using initial RH values of 50% and 80% (for Scenario A) revealed negligible impacts on chemical composition. Thanks for your reminder and we have modified the following description in lines 123~128:

*"For the parcels that stay at the surface, the background temperature was set to 26 ℃ (299 K), and the pressure was set to the standard atmospheric pressure. After updraft, the temperature rapidly dropped to 17 ℃ (290 K) and the pressure became 880 hPa, corresponding to approximately 1.2 km altitude. As the parcel ascended and remained at a high altitude, the emission from ground sources ceased to enter these parcels. To align with the humidity conditions of ascending parcels in CM1, the initial relative humidity was set to 60%."*

1.5  Line 127-130 listed the four scenarios. What is meant by 'high altitude'?

Thank you for your question. The term 'high altitude' used here refers to elevations exceeding 1,000 meters. In PartMC, we set the pressure for parcels remaining near the surface to 1000 hPa. After analyzing the altitude characteristics of parcels ascending to the free troposphere in CM1, we selected 880 hPa (corresponding to ~1200 meters) as representative 'high altitude' conditions. As suggested, we have also added a new table to clarify the environmental parameters at these altitudes, as shown in the response under 1.3.

1.6  Besides, it is not clear about the time dimension of the two set of model simulations, the CM1 and PartMC-MOSAIC. How long does the CM1 simulation last? 6 hours or 9 hours,10 hours or longer? It says PartMC-MOSAIC run for 30 hours, however, some features are shown e.g. concentration of Cl and nitrate increase drastically from the 2nd, 6th and 10th hour for scenarios B, C, and D respectively in Figure 1 (left panel). I wonder what is the relationship between the two 'time axes' of CM1 and PartMC- MOSAIC. Also, it is not clear what is the relationship between the CM1 model output and PartMC-MOSAIC model input.

Thank you for your question. We have clarified these details in the revised manuscript. The CM1 simulation spanned 6 hours, modeling the development of a shallow convective event—a common coastal weather phenomenon driven by daytime solar heating. Thus, its temporal scale does not directly align with the PartMC framework.

In PartMC-MOSAIC, we analyzed the 24-hour aerosol evolution starting at 06:00. To ensure stable results, we initiated the simulation 6 hours earlier (spinup period), excluding this phase from analysis.

The timelines of the two simulations are non-overlapping. The CM1 simulation did not explicitly begin at 06:00 but instead initialized atmospheric conditions prior to shallow convection onset. Shallow convection typically occurs under heterogeneous thermal conditions, frequently observed in coastal regions with daytime solar radiation. Additionally, to compare the cloud-forming properties of aerosol populations under varying emission levels, we designed three parcel ascent scenarios at distinct daytime hours: 08:00, 12:00, and 16:00. These correspond to the 2nd, 6th and 10th hour in the PartMC simulation, when parcels ascend due to shallow cumulus convection. The environmental inputs for these scenarios in PartMC were derived from CM1 parcel trajectory analyses. These conditions are compiled in the new Table 1, appended below Question 1.3.

1.7  How do you deal with exchange of gases and aerosol particles (dilution) between the Lagrangian parcel of PartMC-MOSAIC and the surrounding environment? Is there any model output from CM1 that can guide PartMC- MOSAIC simulations in this regard?

Thank you for your question. The PartMC model incorporates a diffusion algorithm to simulate material exchange between the Lagrangian parcel and its surrounding environment (Riemer et al., 2009). This diffusion module handles both gas and aerosol particle exchanges. Additionally, the algorithm separates horizontal and vertical diffusion coefficients, requiring user-defined inputs for the horizontal dilution coefficient.

In Scenarios B, C, and D, where parcels ascend to higher altitudes, the horizontal diffusion coefficient differs from its surface value. This discrepancy was calibrated using CM1 simulation results. Among the data recorded for each parcel, one parameter, khh, named as the "horizontal eddy diffusivity for scalars", was analyzed. We examined the ratio of khh at the surface to that at higher altitudes in the CM1 results and incorporated it into PartMC. This aspect is briefly discussed in the manuscript (lines 128~129). Additionally, the background gas and aerosol data involved in the dilution process are provided in Table 2 and Table S1, with the relevant modifications described as follows:

*"Subsequently, we converted the data and generated the high-altitude background aerosol used in the simulation. The substances in the ascent parcels would dilute with high-altitude background aerosols and gases. Detailed information on the number concentration and size distribution of the aerosols in different modes can be found in Table 2. The initial aerosol conditions in the parcels contain three modes that are the same as the Ground part because all the parcels stay near the surface during the*

*initialization period. The background concentrations for ground-level and high-altitude gases are listed in Supplementary Information S5.”*

1.8 Line 134, it says that horizontal eddy diffusivity at the high altitude was about one-fifth of the value near the surface. Is there any previous study supporting this assumption?

This discrepancy in the setting originates from CM1 calculations. CM1 employs meteorological simulations to determine the horizontal scalar eddy diffusivity at different altitudes. The parameter configurations for our simulations were based on Siebesma et al. (2003).

The dilution algorithm in PartMC requires user-defined inputs for the horizontal dilution coefficient. For each parcel in CM1, khh, named as the 'horizontal eddy diffusivity for scalars', was recorded with time. We examined the ratio of khh at the surface to that at high altitudes within tracked CM1 parcels and derived this outcome.

1.9 Line 140 and Table 1, how did authors use MERRA-2 reanalysis to derive concentration of aerosol at ground level and at high-altitude? A table showing the aerosol properties derived from MERRA-2 would be helpful, for example, this study is about coastal area, what is the concentration of sea salt aerosol and organic? How high is the 'high altitude'? Since authors also mentioned that vertical variability of aerosol is important, this vertical information is required to be clearly described.

Thank you for your question. The aerosol data at the ground level is from the measurements held in Tsuen Wan, Hong Kong (Wang et al., 2017). Specifically, the observation data contains total mass concentrations of various aerosol species along with their multimodal size distribution parameters. Based on this, we set the ground-level aerosol conditions and applied three-dimensional aerosol reanalysis data from MERRA-2 (the Modern-Era Retrospective Analysis for Research and Applications, Version 2) to help calculate the aerosol concentration at the high altitude. We have revised the description in Section 2.1, which now reads as follows:

*"The aerosol data at the surface was obtained from the observations at a general air quality monitoring station in Hong Kong (Wang and Yu, 2017). The nitrate levels observed in Hong Kong (used in this study) may be higher than other coastal cities due to historical pollution patterns from a decade ago, but the methodology can be adapted to incorporate different observational data from other regions. We extracted the reanalysis aerosol data of Modern-Era Retrospective Analysis for Research and Applications, Version 2 (MERRA-2, DOI: 10.5067/LTVB4GPCOTK2) in the Hong Kong region. The data of the lowest atmospheric layers (from 880 to 1000 hPa) in summer was processed and the vertical proportionality relationships between the ground-level aerosol species and high-altitude species were estimated. Subsequently, we converted the data and generated the high-altitude background aerosol used in the simulation.”*

To meet PartMC's requirement for aerosol inputs, we converted the data and generated the highaltitude background aerosol data used in simulations. These aerosol properties are provided in the Table 2 (line 147). Besides, we have added a new Table S1 in Supplement S5 to provide vertical information of background gas. The high altitude here is 1200-meter high, consistent with the one we mentioned in the scenario design (lines 125~126). We have modified some parts in the methodology and supplementary section. The relevant part is pasted below:

*"Detailed information on the number concentration and size distribution of the aerosols in different modes can be found in Table 2. The initial aerosol conditions in the parcels contain three modes that are the same as the Ground part because all the parcels stay near the surface during the initialization period. The background concentrations for ground-level and high-altitude gases are listed in Supplementary Information S5."*

**Table 2. Aerosol conditions for the emissions** (Riemer et al., 2009b) **and different backgrounds.**

| | N (m$^{-3}$) | $D_{gm}$ (µm) | $\sigma_g$ | Composition by Mass |
|---|---|---|---|---|
| **Emissions** | | | | |
| Meat cooking | $9 \times 10^6$ | 0.0865 | 1.9 | 100% OC |
| Diesel vehicles | $1.6 \times 10^8$ | 0.05 | 1.7 | 30% OC, 70% BC |
| Gasoline vehicles | $5 \times 10^7$ | 0.05 | 1.7 | 80% OC, 20%BC |
| **Ground** | | | | |
| Condensation | $1.26 \times 10^9$ | 0.0816 | 1.6 | 32.6% BC, 48.8% OC, 14% SO$_4$, 5.6% NH$_4$ |
| Droplet | $2.68 \times 10^8$ | 0.14 | 2.1 | 26.4% BC, 23.8% OC, 37.4% SO$_4$, 11.2% NH$_4$, 0.4% Cl, 0.7% NO$_3$ |
| Coarse | $2.3 \times 10^5$ | 2.16 | 1.65 | 11.8% BC, 21% OC, 12.2% SO$_4$, 0.5% NH$_4$, 17.8% Cl, 18.6% NO$_3$, 18.1% Na |
| **High altitude** | | | | |
| Condensation | $3.1 \times 10^8$ | 0.0831 | 1.66 | 15.8% BC, 44.5% OC, 29.8% SO$_4$, 9.9% NH$_4$ |
| Droplet | $6.8 \times 10^7$ | 0.16 | 2.05 | 9.1% BC, 15.5% OC, 57.1% SO$_4$, 17.2% NH$_4$, 0.1% Cl, 1% NO$_3$ |
| Coarse | $6.6 \times 10^4$ | 2.16 | 1.64 | 5.7% BC, 19.1% OC, 25.8% SO$_4$, 1% NH$_4$, 4.4% Cl, 39.6% NO$_3$, 4.4% Na |

**Table S1. The Background Conditions and Emissions for Gases**

| Gas Species | Symbol | Ground Fraction (ppb) | Emissions (nmol m² s⁻¹) | High-altitude Fraction (ppb) |
|---|---|---|---|---|
| Nitric oxide | NO | 0.1 | 31.8 | 0.091 |
| Nitrogen dioxide | $NO_2$ | 1.0 | 1.67 | 0.91 |
| Nitric acid | $HNO_3$ | 1.0 | | 0.91 |
| Ozone | $O_3$ | 50.0 | | 45.5 |
| Hydrogen peroxide | $H_2O_2$ | 1.1 | | 1.001 |
| Carbon monoxide | CO | 21.0 | 291.3 | 19.11 |
| Sulfur dioxide | $SO_2$ | 0.8 | 2.51 | 0.728 |
| Ammonia | $NH_3$ | 0.5 | 6.11 | 0.455 |
| Hydrogen chloride | HCl | 0.7 | | 0.637 |
| Methane | $CH_4$ | 2200.0 | | 2002 |
| Ethane | $C_2H_6$ | 1.0 | | 0.91 |
| Formaldehyde | HCHO | 1.2 | 1.68 | 1.092 |
| Methanol | $CH_3OH$ | 0.12 | 0.28 | 0.1092 |
| Methyl hydroperoxide | $CH_3OOH$ | 0.5 | | 0.455 |
| Acetaldehyde | $ALD_2$ | 1.0 | 0.68 | 0.91 |
| Paraffin carbon | PAR | 2.0 | 96.0 | 1.82 |
| Acetone | AONE | 1.0 | 1.23 | 0.91 |
| Ethene | ETH | 0.2 | 7.2 | 0.182 |
| Terminal olefin carbons | OLET | 0.023 | 2.42 | 0.02093 |
| Internal olefin carbons | OLEI | 0.00031 | 2.42 | 0.0002821 |
| Toluene | TOL | 0.1 | 4.04 | 0.091 |
| Xylene | XYL | 0.1 | 2.41 | 0.091 |
| Lumped organic nitrate | ONIT | 0.1 | | 0.091 |
| Peroxyacetyl nitrate | PAN | 0.8 | | 0.728 |
| Higher organic acid | RCOOH | 0.2 | | 0.182 |
| Higher organic peroxide | ROOH | 0.025 | | 0.02275 |
| Isoprene | ISOP | 0.5 | 0.23 | 0.455 |

| | | | |
|---|---|---|---|
| Alcohols | ANOL | 3.45 | 0.091 |

*"The gaseous species inputs across all simulation scenarios are systematically cataloged in Table S1. These chemical species align with those documented in Zaveri and Peter (1999), while surface background concentrations and emission parameters were adopted from Riemer et al. (2009). For elevated atmospheric conditions, we implemented the ideal gas Clapeyron equation with the following boundary conditions:*

*High altitude: 880 hPa, 290 K*

*Ground level: 1000 hPa, 299 K*

*These conditions enabled derivation of trace gas concentrations at the high altitude of the ascent parcels. Emission rates presented in the table represent temporal averages over emission periods, with actual rates subject to modulation by mixing height variations (Riemer et al., 2009)."*

2.  About the quantification of the aerosol mixing state impact on cloud droplet formation property, presented in section 4 and figure 5, there is no details about how the composition averaging was performed. This is important because the conclusion about impact of mixing state and the error on CCN properties rely on this calculation.

To do the composition averaging, we statistically aggregated the chemical compositions of all particles, assuming each aerosol population exists as a totally internal mixture. While retaining their original size distributions, particles were assigned identical chemical compositions and hygroscopicity parameter. Following your helpful advice, we have added some explanations in this section:

*"Due to measurement limitations on aerosol components, the hygroscopicity of individual particles is often estimated using the average chemical composition of the total aerosol population, which is based on the fully internal mixing assumption. Applying this method we calculate a new CCN concentration $N_{CCN,avg}$, which has a discrepancy compared to the $N_{CCN}$ achieved from the particle-resolved results."*

*"As shown in Figure 5, the CCN activation differences between the composition-averaged and particle-resolved method with the variation of the mixing state index at different environmental supersaturations are investigated. While retaining their original size distributions, particles were assigned identical chemical compositions and hygroscopicity parameters."*

3. Besides, there are already many studies about the relationship between aerosol mixing state and CCN properties of aerosol (the authored also cited some of these studies). What is the uniqueness and novelty of this model study? The authors emphasized their study incorporated shallow cumulus convention model, which is great. However, the authors are expected to explain the importance of such advancement and how does it compare to existing observational studies (even though there is no modeling study of similar kind).

Firstly, our study is the first to integrate the meteorological model CM1 with the aerosol evolution model PartMC. By leveraging CM1, we derived the motion characteristics of air parcels in a coastal area and analyzed the evolution of aerosols from a particle-resolved perspective during parcel transport. This combined approach has built a framework on the aerosol simulations considering the meteorological condition. Although this study focuses on a smaller scale compared to typical meteorological simulations, it lays an important foundation for improving CCN parameterization schemes in large-scale models.

In addition, although PartMC has been extensively applied in previous studies, these studies primarily focused on parcels within the boundary layer. In contrast, this work includes the ascent of parcels into the free troposphere due to meteorological influences, where cloud formation occurs. During this ascent, the environmental conditions of the parcels undergo significant changes (e.g., temperature, pressure, and surrounding gas), which in turn affect aerosol evolution and CCN activation. This aspect is one of the main contributions of our work, as it evaluates aerosol hygroscopicity and cloud-forming potential under realistic cloud-forming conditions.

Besides, this paper reveals that not only the updraft of parcels but also the timing of their ascent will influence the aerosol evolution. The factors affecting aerosol aging processes, including chemical composition, particle size, mixing state and their impact on CCN activation are quantified through systematic analysis of four typical scenarios. The advantages of the particle-resolved model over the composition-averaged method are compared and identified.

Thank you for your suggestions. We have restructured the entire introduction, particularly in the final two paragraphs, to explicitly highlight the novelty (lines 71~84). The revised introduction also analyzes the limitations of traditional observational studies in the third paragraph (lines 44~57). Now the highlight part reads as follows:

*"Although the PartMC-MOSAIC has been extensively applied in previous studies, these studies primarily focused on parcels within the boundary layer, this may limit the accurate representation of environmental conditions when studying the cloud-forming ability of aerosols. Compared to the work mentioned above, Curtis et al. (2017) noted that PartMC, as a zero-dimensional box model, lacks spatial information and has not been integrated into meteorological studies. To address this, they pioneered the coupling of WRF with PartMC, resolving aerosol composition on a per-particle level and integrating aerosol chemistry with meteorology. This represents one of the few studies to combine meteorological modeling with PartMC, which reveals the potential of PartMC in studying aerosol vertical transport. Building on this foundation,*

*our study further explores the application of meteorological simulations coupled with PartMC, advancing the understanding of this approach in aerosol research.*

*Currently, large-scale meteorological simulations generally employ relatively simple aerosol parameterization methods when considering the contribution of CCN to cloud microphysical processes (Hazra et al., 2020; Morrison and Milbrandt, 2011; Thompson and Eidhammer, 2014), such as directly prescribing aerosol activation rates. These simplifications may introduce errors in the prediction of cloud behavior and associated atmospheric processes. To this end, this study integrates the meteorological Cloud Model 1 (CM1) (Bryan and Fritsch, 2002) with the aerosol evolution model PartMC for the first time.”*

4. Line 264, why scenarios B and C differ in aging process? More explanation is expected.

We sincerely appreciate your meticulous attention to detail. We have revised this section in accordance with your suggestions and incorporated additional explanatory text. Specific revisions and additions include the following:

*"In Scenario B, the parcel has been stationed at a high altitude after the 2$^{nd}$ hour and shows a lower aging level compared to others. Conversely, in Scenario C, the ascent process of the parcel has accelerated the aging process, thus shifting the predominant sub-population to a higher $\kappa$ distribution by the 6$^{th}$ hour. Pronounced differences emerged between Scenarios B and C following parcel ascent. This is attributed to the earlier ascent timing in Scenario B, where fewer accumulated precursor gases (e.g., ammonium nitrate precursors) were entrained. Photochemical reactions involving ozone (O$_3$) depend on solar radiation and could affect interactions between nitrogen oxides (NO$_x$). The limited time leads to lower nitric acid levels in Scenario B before ascent compared to Scenario C. As shown in Figure 1 for Scenario A, ground-level pollutant emissions caused rapid nitrate aerosol growth after the 5$^{th}$ hour, preceding the ascent timing in Scenario B. Subsequently, the aerosol populations in Scenarios B and C retained their distinct characteristics but underwent similar aging trajectories.”*

5. Line 286, why some hydrophilic species tend to form on smaller particles? What are those species? More explanation is expected.

This observed phenomenon is derived from the analysis presented in Fig. S4 and Fig. S5 in Supplementary Material S4, which indicates that nitrate components exhibit a preferential tendency to accumulate on smaller-sized particles during the emission process. The term 'hydrophilic species' primarily refers to nitrate components. We have revised the manuscript text to enhance clarity and precision in our description of this aspect.

*"The sensitive impacts of different compositions on the overall $S_c$ are studied in **Supplementary***

*Information S4. As observed in Figures S4 and S5, some hydrophilic species tend to distribute on smaller particles during the aging process, necessitating consideration of the specific mixing state when evaluating the activation properties of particles."*

The modified figures and explanatory captions in the Supplementary section S4 are as follows:

*"Contrary to expectations, particles with higher nitrate amounts exhibit lower capacity to be activated. This suggests that the nitrate formed during the gas-to-particle conversion process may have a greater tendency to adhere to smaller particles, which is further discussed in Fig. S5.*

[Figure]

**Figure S5. The number concentration distribution of the particles with NO3 mass fraction across particle size at 5~8 hours in Scenario A.**

*Taking Scenario A as an example, the nitrate component exhibits significant growth from 5 h to 10 h (as shown in Fig. 1). To illustrate the distribution of nitrate within the aerosol population, the relationship between the $NO_3$ mass fraction and dry diameter at 5 h, 6 h, 7 h, and 8 h for Scenario A is present in Fig. S5. At 5 h, the particles contain negligible amounts of $NO_3$, while as time progresses, a pronounced increase in $NO_3$ mass fraction is observed in particles within the smaller size range. In contrast, the growth of $NO_3$ mass fraction in larger particles demonstrates a more gradual progression. This differential behavior highlights the size-dependent nature of nitrate accumulation in the aerosol population."*

6. Line 360, why at high supersaturation, CCN activation error becomes larger for some scenarios? More explanation is expected.

Thank you for your suggestion. We have expanded the explanatory text in Section 4. The relevant part including the added explanations is pasted below:

*"Specifically, in Scenario A, this phenomenon is more pronounced, and smaller $\chi$ values correspond to larger error values. This is because, under higher supersaturation conditions, smaller particles within the population exhibit a greater likelihood of activation. In Scenario A, these smaller particles are primarily composed of freshly emitted pollutants with lower hygroscopicity. Consequently, even under elevated supersaturation levels, this subpopulation remains inactivated in the real world. PartMC can accurately capture the mixing state reflected by a smaller mixing state index $\chi$, and correctly calculate the critical supersaturation of these particles. In contrast, the composition-averaged method overestimates the*

*hygroscopicity of these small, freshly emitted particles. This overestimation leads to an elevated critical supersaturation, making these particles more likely to be assumed as CCN under higher supersaturation conditions. As a result, the CCN predictions based on the internal mixing assumption become more sensitive to variations in environmental supersaturation."*

7. In figure 4b, why is there a peak for scenarios B, C and D at 2nd and 6th hour?

We appreciate your question regarding the mixing state index. The increase in the mixing state index signifies a tendency towards internal mixing (aging) within the aerosol population. In Scenario B, as the air parcel ascends from the surface, the emission process generating externally mixed particles ceases, while the aging process continues. This could lead to a transient increase in $\chi$, followed by a subsequent decline in the mixing state index as photochemical reactions gradually elevate NO₃ concentrations. When all nitrogen from the initial emission process is converted to NO₃, the mixing state index increases over time until reaching a stable equilibrium.

In Scenario C, despite the cessation of emissions, the sudden generation of substantial NO₃ during the ascent process results in the aerosol population exhibiting a more externally mixed state overall. Scenario D, prior to ascent, shares the same initial conditions as Scenario A, with the peak occurring around 6 h, corresponding to the state of maximal mixing within the system.

**Specific comments**

Apart from the general comments above, there are some specific questions or items to be clarified throughout the manuscript.

1. Section 4, (p.14) There is no details about the calculation of 'chi', the mixing state index. In the figure 4a, the $D_\alpha$ and $D_\gamma$ lie between 1.4 to 2.0, why is the range so small? There are so many chemical species listed in lines 160-164.

Thank you for your question. In Section 4, we have expanded the explanation of the mixing state index defined using Shannon entropy.

The average per-particle species diversity Dα and the bulk population species diversity Dγ are based on the definitions provided by Riemer and West (2013). Theoretically, these indices can range from 1 to A, where the maximum value A represents the number of distinct aerosol species. In practical applications, the value of A can vary depending on how 'species' are defined and grouped, rather than being limited to real chemical species. For instance, the value A can be defined based on species groups (e.g., Dickau et al., 2016, Ching et al., 2017). In our study, since we are concerned with CCN properties, the species are categorized into three groups based on their hygroscopicity:

1. Low-hygroscopicity species: black carbon (BC) and primary organic aerosol (OC).

2.  High-hygroscopicity species: sulfate ($SO_4$), nitrate ($NO_3$), ammonium ($NH_4$), chloride (Cl), and sodium (Na).

3.  Intermediate-hygroscopicity species: secondary organic aerosol (SOA).

As a result, the maximum value of Dα and Dγ equals 3, instead of the total number of individual chemical species. The observed range of Dα and Dγ in this study ranges from 1.4 to 2, thus we have incorporated a broken axis in Figure 4a for better visual resolution.

We have modified the description to avoid misunderstanding. The revised part now reads:

*"The mixing state index (χ) is defined by Riemer and West (2013) to quantify the aerosol mixing state, which is calculated by,*

$$\chi = \frac{D_\alpha - 1}{D_\gamma - 1} \tag{4}$$

*where $D_\alpha$ represents the average per-particle species diversity, $D_\gamma$ represents the bulk population species diversity. Given the focus on analyzing CCN properties, the aerosol species are classified into three distinct categories based on their hygroscopicity: low-hygroscopicity species (BC, OC), high-hygroscopicity species ($SO_4$, $NO_3$, $NH_4$, Cl, Na), and intermediate-hygroscopicity species (SOA). As a result, the value of $D_\alpha$ and $D_\gamma$ ranges from 1 to 3."*

References for this question:

Riemer, N. and West, M.: Quantifying aerosol mixing state with entropy and diversity measures, Atmospheric Chemistry and Physics, 13, 11423–11439, https://doi.org/10.5194/acp-13-11423-2013, 2013.

Ching, J., Fast, J., West, M., & Riemer, N. (2017). Metrics to quantify the importance of mixing state for CCN activity. Atmospheric Chemistry and Physics, 17(12), 7445–7458. https://doi.org/10.5194/acp-17-7445-2017

Dickau, M., Olfert, J., Stettler, M. E. J., Boies, A., Momenimovahed, A., Thomson, K., Smallwood, G., & Johnson, M. (2016). Methodology for quantifying the volatile mixing state of an aerosol. Aerosol Science and Technology. https://www.tandfonline.com/doi/abs/10.1080/02786826.2016.1185509

2.  What is the definition of error in Figure 5? Does positive sign mean overestimation or underestimation of CCN by composition averaging? How about negative sign?

Thank you for pointing out the lack of clarity in our explanation. We have incorporated a mathematical formula to define the error term in the revised manuscript.

In the original Figure 5, the error was quantified using the following expression: (CCN concentration calculated via composition averaging - CCN concentration from the PartMC model) / total aerosol

concentration. The positive sign indicates an overestimation of CCN concentration through composition averaging, whereas the negative value signifies an underestimation resulting from the averaging method.

Following another reviewer's suggestion, the new y-axis in Fig. 5 employs a percentage scale and represents the absolute value. Hence, we have revised both the figure and the corresponding discussion. The newly added formula and modified figure are pasted below:

*"Due to measurement limitations on aerosol components, the hygroscopicity of individual particles is often estimated using the average chemical composition of the total aerosol population, which is based on the fully internal mixing assumption. Applying this method we calculate a new CCN concentration $N_{CCN,avg}$, which has a discrepancy compared to the $N_{CCN}$ achieved from the particle-resolved results. The total particle concentration is defined as $N_{total}$, and the difference can be calculated by the following equation,*

$$Difference\ in\ CCN\ Activation\ Ratio = \frac{\left|N_{CCN,avg} - N_{CCN}\right|}{N_{total}} \qquad (5)$$

*The internal mixing assumption simplifies the calculations and the measurements and is widely used to investigate CCN activation. However, the real-world mixing state of aerosols could largely deviate from this assumption, which means the actual $\chi$ value is lower than 1. Since only particle-resolved methods accurately represent the true mixing state of aerosols, we use the CCN activation error to represent the difference between the composition-averaged and particle-resolved methods.*

[Figure]

**Figure 5. The difference in CCN activation between composition-averaged and particle-resolved results with the mixing state index $\chi$ at various environmental supersaturations for four scenarios. The scatters show the results at ten-minute intervals.**

*As shown in Figure 5, the CCN activation differences between the composition-averaged and particle-resolved method with the variation of the mixing state index at different environmental supersaturations are investigated. While retaining their original size distributions, particles were assigned identical chemical compositions and hygroscopicity parameters."*

---

## Author Response (AR2)

**Response to referee comments**

Journal: Atmospheric Chemistry and Physics

Manuscript ID: Preprint egusphere-2024-3581

Title: Exploring the Aerosol Activation Properties in Coastal Shallow Convection Using Cloud and Particle-resolving Models

Authors: Ge Yu, Yueya Wang, Zhe Wang, Xiaoming Shi

We thank the reviewers for the comments and suggestions, following which we revised the manuscript. Below are our responses to the reviewers' comments. The text in black is the original comments from reviewers, and our **responses** are the **text in blue**. Some text from the revised manuscript is quoted in this response letter for the reviewers' convenience, and the *quoted text* is in *italic font*.

**Referee 1:**

The authors have addressed the reviewers' questions well, and the paper is improved in many ways. There are a few issues left that require another round of revisions. They are minor in nature, so I suggest minor revisions.

1. Motivation of this study: in the introduction, the authors now say: "Currently, large-scale meteorological simulations generally employ relatively simple aerosol parameterization methods when considering the contribution of CCN to cloud microphysical processes (Thompson and Eidhammer, 2014; Morrison and Milbrandt, 2011; Hazra et al., 2020), such as directly prescribing aerosol activation rates. These simplifications may introduce errors in the prediction of cloud behavior and associated atmospheric processes. To this end, this study integrates the meteorological Cloud Model 1 (CM1) (Bryan and Fritsch, 2002) with the aerosol evolution model PartMC-MOSAIC for the first time." I agree with you that CCN in large-scale models are represented overly simplified, but this paper does not directly address the impact of these simplifications. Could you add something to the paper that addresses the question how directly prescribing aerosol activation rates would introduce error? (This is even more simplified than using composition-averaged information, which is what the paper does investigate.) This would be very useful for large-scale modelers.

Thank you for your suggestions. We have added further explanations regarding the error mechanisms associated with simplified parameterizations.

The phrase "such as directly prescribing aerosol activation rates" refers to a method for estimating the relationship between CCN number concentration and ambient supersaturation. A widely used approach is the power-law parameterization proposed by Twomey (1959), expressed as $N_{CCN} = C \cdot S^k$, where S

represents supersaturation, and C and k are coefficients derived from observed CCN characteristics, typically prescribed as parameters in simulations. In observational data, particles that do not reach the critical activation diameter are sometimes misidentified as CCN, introducing errors in real-time CCN estimated by the formula. Due to its simplicity, this approach remains widely used to date. However, it undeniably yields coarse CCN estimates that deviate from real-world conditions. This deviation further impacts the calculation of critical properties such as cloud droplet number concentration and droplet radius, which may affect the treatment of cloud water conversion processes in models (Fan et al., 2012). Therefore, improving the accuracy of CCN concentration estimates is essential to reduce model uncertainties. When simulating a case of shallow cumulus clouds, Wang et al. (2025) revised the basic relationship, revealing that the uncorrected power-law parameterization for CCN suppresses precipitation formation and overestimates cloud radiative cooling. Notably, the effectiveness of the correction decreases under high aerosol loading conditions. Additionally, Hazra et al. (2020) compared different microphysics schemes within the WRF model for the same case, highlighting significant variability in results. These simplifications in CCN estimation can lead to errors in hydrometeor predictions, thereby affecting the accuracy of cloud behavior and associated atmospheric processes.

We have expanded the description in this section to improve clarity as suggested. The updated content is provided below.

*'Currently, large-scale meteorological simulations typically employ simplified aerosol parameterization methods to represent the contribution of cloud condensation nuclei (CCN) to cloud microphysical processes (Thompson and Eidhammer, 2014; Morrison and Milbrandt, 2011). For instance, a common approach directly relates CCN concentration to ambient supersaturation through power-law parameterization. As proposed by Twomey (1959), the function is expressed as $N_{CCN} = C \cdot S^k$, where $S$ represents supersaturation, and $C$ and $k$ are coefficients derived from observed CCN characteristics, typically prescribed as parameters in simulations. Due to its simplicity, this approach remains widely used in many microphysical schemes (Hong et al., 2010; Mansell et al., 2010; Morrison et al., 2009). However, it undeniably yields coarse CCN estimates that deviate from real-world conditions. This deviation further impacts the calculation of critical properties such as cloud droplet number and droplet radius, which may affect the cloud-rain conversion processes in models (Fan et al., 2012). When simulating a case of shallow cumulus clouds, Wang et al. (2025) corrected the basic power-law, revealing that the uncorrected power-law parameterization for CCN suppresses precipitation formation and overestimates cloud radiative cooling. Notably, the effectiveness of the correction decreases under high aerosol loading conditions. Comparative studies, such as Hazra et al. (2020), further demonstrate significant variability in results across different microphysical schemes within the WRF model for the same case. The simplifications can affect CCN concentrations, potentially introducing errors in hydrometeor predictions and impacting cloud behavior and related atmospheric processes. Therefore, the accuracy of CCN concentration is essential to reduce model uncertainties. Given the distinct characteristics of convective cloud processes across different vertical regions, the influence of vertical CCN distribution differences on precipitation is also a*

*promising topic. Improving the estimation of CCN variation in cloud-forming locations is crucial for enhancing cloud microphysical schemes. To this end, this study integrates the meteorological Cloud Model 1 (CM1) (Bryan and Fritsch, 2002) with the aerosol evolution model PartMC-MOSAIC for the first time. The CM1 is applied to initially investigate the movement characteristics of air parcels under a shallow cumulus convection condition. Key parameters obtained from the CM1 experiments and observations are then introduced into the PartMC-MOSAIC model to investigate the evolution of aerosol populations across different air parcel scenarios. By analyzing key indicators such as hygroscopicity, critical supersaturation, and mixing state index, this study explores the cloud-forming abilities of aerosols within different parcels.'*

2.    Re: Comment 3: Figure S1: Average over the entire x-y plane not meaningful. Suggest to only average over the core of the cloud? In general, it'd be helpful to visualize the parcel trajectories better, i.e. include a 3D plot of a few example trajectories. I'm not sure how feasible this is, but it would illustrate the purpose (and novelty) of this study well.

Thanks for your comments. Comment 3 raised questions about the cloud water and ice content subplot: the horizontally averaged qc values in the figure might be too small and potentially confusing to readers. As suggested, we have added a subplot that shows the average of cloud water where its value exceeds $10^{-5}$ kg/kg (i.e., the cloud core average). The revised figure and captions are presented below.

[Figure]

**Figure S1. Temporal evolution of horizontal-averaged (a) potential temperature, (b) water vapor mixing ratio, and (c) cloud water and ice mixing ratio with the height in the preliminary CM1 experiment. Panel (d) depicts the conditional mean cloud water and ice mixing ratio over the cloud core (defined as regions where the cloud water mixing ratio exceeds $10^{-5}$ kg/kg).**

We have also added a 3D plot featuring representative trajectory examples in this section. The plot includes both rapidly ascending air parcels (different ascent timings) and a parcel that remains predominantly within the boundary layer. The corresponding figure and modified descriptions are presented below.

[Figure]

**Figure S2. Three-dimensional trajectories of three representative air parcels simulated by the CM1 model. The x and y positions of the parcels have been adjusted to account for the horizontal periodic boundary conditions of the simulation domain, ensuring continuous trajectories across domain boundaries.**

*'The presentation of Fig. S1 is to illustrate the potential cloud formation heights under the typical shallow cumulus convection conditions. As stated in the main text, approximately one-third of the air parcels ascend to altitudes exceeding 1000 meters, where clouds could form. Figure S2 shows three representative parcel trajectories simulated using CM1. In this case, there is an initial background wind speed of approximately 8 m/s in the -x direction. The x and y positions of the air parcels have been adjusted to eliminate discontinuities arising from the horizontal periodic boundary conditions of the simulation domain. The trajectory plot includes both rapidly ascending parcels (with varying ascent timings) and a parcel that remains predominantly within the boundary layer.'*

3. Re: Comment 9: Thanks for adding information about the MERRA-2 dataset. However, it is still not clear to me how the mass concentrations of sulfate, black carbon, organic carbon and sea salt from MERRA are converted into the data that PartMC can use. I believe, you needed to apply some assumptions about size distributions and mixing state. What are these assumptions?

We thank the reviewer for the comments. Based on ground observations, size distributions were reported and aerosol particles were classified into three modes. Lognormal size distribution for particles and internal mixing assumption in every mode were adopted to ensure compatibility with the model.

After calculating the new mass concentrations of species at higher altitudes, we applied the same size distribution and internal mixing assumptions for the three modes as those used at ground level (as described in lines 150 ~ 156). That is, these assumptions and transformation methods were identical for both high-altitude and surface aerosols. As reminded by the comment, we have also revised some descriptions. The relevant content is pasted below:

*'For cloud-altitude aerosol background conditions, we extracted the reanalysis aerosol data of Modern-Era Retrospective Analysis for Research and Applications, Version 2 (MERRA-2, DOI: 10.5067/LTVB4GPCOTK2) in the Hong Kong region. The data of the lowest atmospheric layers (from 880 to 1000 hPa) in summer was processed and the vertical proportionality relationships between the ground-level aerosol species and cloud-altitude species were estimated. Subsequent calculations determined the mass concentrations of different species at elevated altitudes. The same size distribution and mixing state assumptions as those used for ground-level conditions were applied, and the cloud-altitude background aerosol for the simulation was ultimately generated.'*

4. Table S1: Instead of "ground fraction"/ "high-altitude fraction", I suggest using "mixing ratio at ground level" and "mixing ratio at elevated altitude" (I would not consider 880 hPa as "high-altitude")

We have revised the descriptions as suggested. Additionally, a detailed explanation of more modifications made to Table S1 has been provided in our response to Question 5.

Following the reviewer's suggestions, we have replaced the term "high altitude" with either "cloud altitude" or "elevated altitude" throughout the manuscript to enhance the precision and clarity of the expression.

5.    Furthermore, the following sentences are unclear (in section S5): "For elevated atmospheric conditions, we implemented the ideal gas Clapeyron equation with the following boundary conditions". There is the ideal gas law, and there is the Clausius-Clapeyron equation. I assume you mean the ideal gas law here, but how did you use it in this context to obtain the numbers in the column for "high-altitude fraction"?

Yes, we quite mean the ideal gas law here. Thank you for raising this issue. Initially, we mistakenly assumed the required 'gas concentration' in the model to be mass volume fraction. Thus, we applied the ideal gas law and calculated the ratio of mass concentration between the elevated altitude and the surface, which was incorrect for the input conditions. Upon revisiting the model's framework, we recognized that it employs the mixing ratio (unit: ppb) for gas species. Reapplying the ideal gas law demonstrated that the volume mixing ratios of the trace gases remain constant with altitude.

To fix this problem, we have revised the elevated-altitude background gas conditions in Scenarios B, C, and D, which marginally influence diffusion processes after ascent. After re-running the simulations and updating the data analyses, we have updated all relevant figures and descriptions accordingly. While the revised figures show minor differences (the comparison is visible in the tracked-changes PDF), these adjustments do not affect the study's conclusions, as the patterns and quantitative relationships remain virtually unchanged and consistent with our previous results. Additionally, we updated Supplementary S5, including the table stating the background gas conditions and associated explanations. The modified Table S1 and descriptions are presented below:

**Table S1. The Background and Emission Conditions for Gases**

| Gas Species | Symbol | Mixing Ratio of the Background Gas (ppb) | Emissions (nmol m² s⁻¹) |
|---|---|---|---|
| Nitric oxide | $NO$ | 0.1 | 31.8 |
| Nitrogen dioxide | $NO_2$ | 1.0 | 1.67 |
| Nitric acid | $HNO_3$ | 1.0 | |
| Ozone | $O_3$ | 50.0 | |
| Hydrogen peroxide | $H_2O_2$ | 1.1 | |
| Carbon monoxide | $CO$ | 21.0 | 291.3 |
| Sulfur dioxide | $SO_2$ | 0.8 | 2.51 |
| Ammonia | $NH_3$ | 0.5 | 6.11 |
| Hydrogen chloride | $HCl$ | 0.7 | |
| Methane | $CH_4$ | 2200.0 | |
| Ethane | $C_2H_6$ | 1.0 | |
| Formaldehyde | $HCHO$ | 1.2 | 1.68 |
| Methanol | $CH_3OH$ | 0.12 | 0.28 |
| Methyl hydroperoxide | $CH_3OOH$ | 0.5 | |
| Acetaldehyde | $ALD_2$ | 1.0 | 0.68 |
| Paraffin carbon | $PAR$ | 2.0 | 96.0 |

| | | | |
|---|---|---|---|
| Acetone | AONE | 1.0 | 1.23 |
| Ethene | ETH | 0.2 | 7.2 |
| Terminal olefin carbons | OLET | 0.023 | 2.42 |
| Internal olefin carbons | OLEI | 0.00031 | 2.42 |
| Toluene | TOL | 0.1 | 4.04 |
| Xylene | XYL | 0.1 | 2.41 |
| Lumped organic nitrate | ONIT | 0.1 | |
| Peroxyacetyl nitrate | PAN | 0.8 | |
| Higher organic acid | RCOOH | 0.2 | |
| Higher organic peroxide | ROOH | 0.025 | |
| Isoprene | ISOP | 0.5 | 0.23 |
| Alcohols | ANOL | | 3.45 |

*'The gaseous species inputs across all simulation scenarios are systematically cataloged in Table S1. These chemical species align with those documented in Zaveri and Peter (1999), while surface background concentrations and emission parameters were adopted from Riemer et al. (2009). For the background gas concentration at the elevated altitude, we assume no additional substances are introduced into the system. Based on the ideal gas law, the volume mixing ratios of these background trace gases remain constant between the surface and elevated altitudes.'*

6.    Equation (5): Why is the denominator the total aerosol concentration? Usually when defining the error in a quantity, one would divide by the reference value (N_CCN in this case).

Thank you for your question. Dividing N_CCN by the total aerosol concentration allows us to calculate the CCN activation ratio. We aimed to directly compare differences in the CCN activation ratio across different scenarios after performing composition averaging.

Additionally, as discussed in Section 4.2 and illustrated in Figure 6, N_CCN may occasionally equal zero in smaller size groups, which introduces complications when used as a denominator. To maintain consistency in error calculations throughout Chapter 4, we adopted the formula specified in Equation (5).

7.    New figure 6: I suggest labelling the x-axis with the size ranges of the bins that the particles were grouped into.

We have revised Figure 6 according to the suggestions provided. The modified figure and its captions are presented below.

[Figure]

**Figure 6. The difference in CCN activation between composition-averaged and particle-resolved approaches within 5 logarithmic size bins ($10^{-3}$ ~ $10^2$ μm) at various environmental supersaturations for four scenarios. Boxplots show 25th–75th percentiles (with median line); data comes from ten-minute intervals over the simulation.**

8. Nomenclature: The authors use "hydrophilic" and "hygroscopic" interchangeably. The two terms relate to how substances interact with water, but they're not exactly interchangeable. When referring to substances like sulfate, ammonium, and nitrate, I suggest calling them hygroscopic species, because this is directly related to water uptake and CCN activity.

We appreciate the reviewer's useful comments. We have replaced all instances of 'hydrophilic' with 'hygroscopic' in the manuscript and figures. For convenience, the modified Figure 1 is pasted below.

[Figure]

**Figure 1. Mass concentration variations over time of hygroscopic and carbonaceous components in Scenario A, B, C, and D. For Scenario A: the parcel remains near the ground; B, C and D: the parcel ascends after 2 hours, 6 hours and 10 hours respectively.**

9. Choice of title: I think the title is too generic. In my opinion, this paper is really about different pathways of aerosol aging, depending on how quickly the air parcel is cut off from the emissions at the ground. Suggest something like: "Modeling Aerosol Aging and Cloud Activation During Air Parcel Ascent in Coastal Shallow Convection"

Thank you for your suggestions. After careful consideration, we agree that the title should be more specific, as you noted, to better reflect the shallow convection condition. Additionally, we aim to highlight that this work utilizes the cloud model and the PartMC-MOSAIC model. Accordingly, we have revised the title to: "Exploring Aerosol Activation Properties in Coastal Shallow Convection Using Cloud and Particle-Resolving Models."

10. Last sentence: "Additionally, the mixing state index has the potential to be accounted for in future climate models as it is still challenging to represent some microscale aerosol-cloud interactions and reduce the microphysical uncertainties in large-scale models." I'm not clear on how this is supposed to work.

Thanks for your comments. The mixing state index has been implemented in several climate models. Shen et al.(2024) divided an aerosol mode (previously assumed as internal mixing) into two parts in the Community Atmosphere Model Version 6 (CAM6): one containing BC and one without BC. By

prescribing the ratio of these two parts, the real-time mixing state index χ, can be calculated. They employed a machine learning model to predict the χ in advance and continuously adjusted the mentioned proportion in CAM6 to align the real-time calculated χ with the machine learning predictions. Through this χ-based adjustment process, the new predicted global average ratio of BC-containing particles to BC-free particles was found to be closer to observations (the error is 52% lower than that from traditional CAM6 simulations).

Current global climate models still exhibit biases in representing aerosol mixing states, and improvements in this area could help reduce uncertainties in aerosol-cloud interactions. Further research on the impact of aerosol mixing states on aerosol activation in convective clouds is also of great interest (Shan et al., 2021; Wang et al., 2013).

To enhance the manuscript clarity, we have revised the original text as follows:

*'Additionally, the mixing state index can be incorporated into future climate models to better constrain aerosol-cloud interactions. Shen et al.(2024) developed a machine learning model coupled in the Community Atmosphere Model Version 6 (CAM6) that can online correct biases in the mixing states. The χ-guided adjustments provide a more accurate treatment of BC-related processes, including mixing state and coating status. In the future, χ has the potential to dynamically adjust other critical processes, offering a pathway to reduce microphysical uncertainties without requiring explicit microscale resolution.'*

New mentioned references:

Shan, Y., Liu, X., Lin, L., Ke, Z., & Lu, Z. (2021). An Improved Representation of Aerosol Wet Removal by Deep Convection and Impacts on Simulated Aerosol Vertical Profiles. *Journal of Geophysical Research: Atmospheres*, *126*(13), e2020JD034173. https://doi.org/10.1029/2020JD034173

Shen, W., Wang, M., Riemer, N., Zheng, Z., Liu, Y., and Dong, X.: Improving BC Mixing State and CCN Activity Representation With Machine Learning in the Community Atmosphere Model Version 6 (CAM6), Journal of Advances in Modeling Earth Systems, 16, e2023MS003889, https://doi.org/10.1029/2023MS003889, 2024.

Wang, H., Easter, R. C., Rasch, P. J., Wang, M., Liu, X., Ghan, S. J., Qian, Y., Yoon, J.-H., Ma, P.-L., & Vinoj, V. (2013). Sensitivity of remote aerosol distributions to representation of cloud–aerosol interactions in a global climate model. *Geoscientific Model Development*, *6*(3), 765–782. https://doi.org/10.5194/gmd-6-765-2013

**Referee 2:**

1.  I would recommend the authors to provide more details about the exchange (of aerosol species an gases) between the parcel and the environment, i.e. due to diffusion in both vertical and horizontal directions (related to my original comments 1.7 and 1.8), especially the numerical values of dilution coefficients input to PartMC-MOSAIC. Besides, the times series of eddy diffusivity of scenarios A-D would be helpful for readers to understand the impact of parcel motion (driven by meteorology) on aging of aerosol particles.

Thank you for your suggestions. In subsection 2.3, we have expanded the explanation of the PartMC model's diffusion algorithm, including separate treatments of vertical and horizontal components. For the vertical diffusion influenced by diurnal mixing height variation and the surface-level horizontal diffusion coefficient, we adopted the specific values from Riemer et al. (2009). The ground coefficient equals $1.5 \times 10^{-5}$ s$^{-1}$ and the elevated- altitude one equals $3 \times 10^{-6}$ s$^{-1}$. The modified description is as follows:

*'The PartMC model incorporates a diffusion algorithm to simulate material exchange between the Lagrangian parcel and its surrounding environment, handling both gas and aerosol particle exchanges. Additionally, the algorithm separates horizontal and vertical diffusion coefficients. The vertical diffusion rate depends on both the magnitude and the variation rate of mixing height, with an increase in the mixing height leading to positive values. The horizontal coefficient requires user-defined input and represents the effects of horizontal turbulent diffusion. The diurnal variation of the mixing height and the surface-level horizontal coefficient in this study are consistent with Riemer et al. (2009). When the parcel is at ground level, the horizontal diffusion coefficient is set as $1.5 \times 10^{-5}$ s$^{-1}$. After the parcel rises, the value of mixing height is constant to achieve zero vertical diffusion rate, simulating free tropospheric conditions. The horizontal coefficient at the elevated altitude is set as $3 \times 10^{-6}$ s$^{-1}$, corresponding to the one-fifth ratio relative to the surface coefficient as described in subsection 2.1. The temporal evolution of horizontal coefficients for Scenarios A~D is documented in* **Supplementary Information S5**.*'*

Besides, the time series of the horizontal diffusion coefficients for Scenarios A-D have been added to Table S2 in the Supplementary Information S5. The specific contents are as follows:

**Table S2. The Time Series of the Horizontal Dilution Coefficients in Different Scenarios**

| Scenario | Time (h) | Coefficient (s$^{-1}$) |
|----------|----------|------------------------|
| A | 0 ~ 24 | $1.5 \times 10^{-5}$ |
| B | 0 ~ 2 | $1.5 \times 10^{-5}$ |
| B | 2 ~ 24 | $3 \times 10^{-6}$ |
| C | 0 ~ 6 | $1.5 \times 10^{-5}$ |

| | | |
|---|---|---|
| C | 6 ~ 24 | $3 \times 10^{-6}$ |
| D | 0 ~ 10 | $1.5 \times 10^{-5}$ |
| D | 10 ~ 24 | $3 \times 10^{-6}$ |

*'In Table S2, the displayed time aligns with the temporal scale used for result analysis in the main text, excluding the 6-hour spin-up period. In the 6-hour spin-up PartMC-MOSAIC simulation, the horizontal diffusion coefficients were maintained at $1.5 \times 10^{-5} \, s^{-1}$ since the parcels were near the surface throughout the initialization phase.'*